# A reinforced lunar dynamo recorded by Chang'e-6 farside basalt

Shuhui Cai[1,2✉], Kaixian Qi[1,2], Saihong Yang[3], Jie Fang[1,2], Pingyuan Shi[1,2], Zhongshan Shen[1], Min Zhang[4], Huafeng Qin[1,2], Chi Zhang[4], Xiaoguang Li[1], Fangfang Chen[1,2], Yi Chen[1,2], Jinhua Li[2,4], Huaiyu He[1,2], Chenglong Deng[1,2], Chunlai Li[3], Yongxin Pan[2,4] & Rixiang Zhu[1]

The evolution of the lunar dynamo is essential for deciphering the deep interior structure, thermal history and surface environment of the Moon[1–4]. Previous palaeomagnetic investigations on samples returned from the nearside of the Moon have established the general variation of the lunar magnetic field[5–7]. However, limited spatial and temporal palaeomagnetic constraints leave the evolution of the lunar dynamo ambiguous. The Chang'e-6 mission returned the first farside basalts dated at about 2.8 billion years ago (Ga) (refs. 8,9), offering an opportunity to investigate a critical spatiotemporal gap in the evolution of the global lunar dynamo. Here we report palaeointensities (around 5–21 μT) recovered from the Chang'e-6 basalts, providing the first constraint on the magnetic field from the lunar farside and a critical anchor within the large gap between 3 Ga and 2 Ga. These results record a rebound of the field strength after its previous sharp decline of around 3.1 Ga, which attests to an active lunar dynamo at about 2.8 Ga in the mid-early stage and argues against the suggestion that the lunar dynamo may have remained in a low-energy state after 3 Ga until its demise. The results indicate that the lunar dynamo was probably driven by either a basal magma ocean or a precession, supplemented by other mechanisms such as core crystallization.

Large-scale crustal magnetizations distributed across much of the surface of the Moon (Fig. 1) demonstrate that there were once magnetic sources, such as lunar dynamo, and perhaps impact-generated fields that magnetized the lunar crust[6,10]. Palaeomagnetic study of the returned samples from the Apollo and Chang'e-5 missions has established our general knowledge of the evolution of the lunar magnetic field. The palaeointensity data available now suggest that the Moon once possessed an active dynamo from 4.2 Ga to 3.5 Ga, which first dropped by one order of magnitude around 3.1 Ga and maintained a low field of several μT until a second decline between around 1.5 Ga and 1 Ga and finally ceased entirely at some point after about 1 Ga (refs. 2,7,11–17). However, published data are mainly concentrated before about 3 Ga with few constraints thereafter. The Chang'e-5 basalts recently provided an important new anchor point at around 2 Ga (ref. 17). Nonetheless, the still sparsely sampled intermediate evolution of the lunar magnetic field remains relatively poorly constrained. Furthermore, all previous data are derived from samples collected from the nearside of the Moon. At present, there are no direct constraints on the palaeomagnetic field from the farside of the Moon. Therefore, long-standing issues such as the lifetime, geometry and driving mechanisms of the lunar magnetic field remain greatly debated, in which such ambiguity leads to unresolved concerns about the inconsistency between orbital and sample-based measurements data[18] and even permits these contrary perspectives as arguing against the existence of a long-lived lunar core dynamo altogether[19,20].

The Chang'e-6 mission returned the first samples collected from the lunar farside from the southern Apollo crater inside the South Pole–Aitken basin (41.64 °S, 153.99 °W) (ref. 21) (Fig. 1). The samples used in this study (CE6C0300YJFM002) were scooped from the lunar surface and allocated by the China National Space Administration. A total of 4-mm-scale basalt clasts (CE6C0000YJYX211, 038, 344 and 392) dated at about 2.8 Ga by the Pb–Pb dating method[8] were used for palaeointensity, rock magnetic and microscopy analyses (Fig. 2, Methods and Extended Data Table 1). Thus, given the farside location and unique mid-stage age of these samples, palaeomagnetic study of the Chang'e-6 basalt holds great potential for providing important constraints on the uncertain spatiotemporal evolution of the global lunar dynamo, and thus for resolving the existing controversies over the state and power source of the lunar dynamo.

## Palaeomagnetism of the Chang'e-6 basalt

To recover the palaeointensity estimates of the lunar magnetic field for the Chang'e-6 basalt clasts, non-thermal palaeointensity techniques, including anhysteretic remanent magnetization (ARM) and isothermal remanent magnetization (IRM) correction methods[22,23], were used (Methods and Supplementary Discussion 1). Natural remanent magnetizations (NRMs) of the four studied basalt clasts vary from $3.22 \times 10^{-10}$ Am$^2$ to $3.22 \times 10^{-9}$ Am$^2$ after viscous remanent magnetization

[1]State Key Laboratory of Lithospheric and Environmental Coevolution, Institute of Geology and Geophysics, Chinese Academy of Sciences, Beijing, China. [2]College of Earth and Planetary Sciences, University of Chinese Academy of Sciences, Beijing, China. [3]Key Laboratory of Lunar and Deep Space Exploration, National Astronomical Observatories, Chinese Academy of Sciences, Beijing, China. [4]Key Laboratory of Earth and Planetary Physics, Institute of Geology and Geophysics, Chinese Academy of Sciences, Beijing, China. ✉e-mail: caishuhui@mail.iggcas.ac.cn

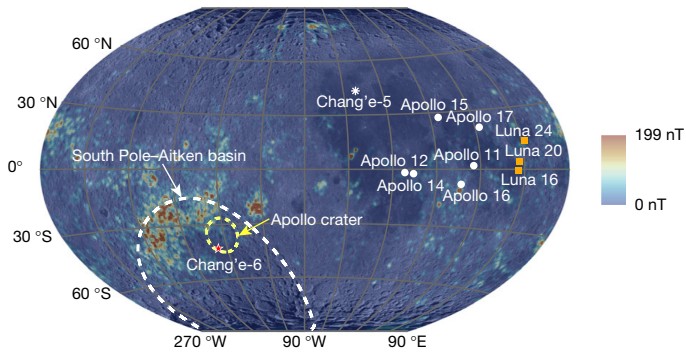

**Fig. 1 | Magnetic anomalies on the lunar surface and the landing sites of lunar exploration missions.** Magnetic anomaly data are calculated with the lunar magnetic field model of ref. 25. A spherical surface of 1,737 km from the centre of the Moon was used as the lunar surface. The Winkel Tripel projection, centred on the lunar nearside–farside boundary at 90 °W, was used for the map. The landing sites of the lunar missions are indicated. The South Pole–Aitken basin and Apollo crater are shown as dashed white and yellow lines, respectively. Different symbols represent various lunar exploration missions.

(VRM) decay. During alternating field (AF) demagnetization, the NRMs of the samples display at least three components (Fig. 3 and Extended Data Figs. 1–3). The low coercivity magnetization components persist until 4–10 mT, which may be attributed to the VRM overprints acquired when exposed to the magnetic field of Earth on sample return. The medium coercivity components vary among the samples, lasting until 24–50 mT. Some samples include complex medium coercivity directions, including two opposite directions, such as from 36 mT to 56 mT for sample 038 and from 11 mT to 36 mT for sample 392. The high coercivity components generally persist from ≥26 mT until 100–150 mT. Samples 211 and 038 exhibit relatively stable high coercivities decaying to the origin with maximum angular deviations (MADs) larger than deviation angles (DANGs), whereas the high coercivity of sample 344 shows a large scatter but could also be treated as origin-trending as its MAD exceeds its DANG (Extended Data Table 1).

Three of the four sample clasts (only excluding 392) yield high coercivities considered characteristic remanent magnetizations; their high coercivities were used to calculate the palaeointensities. Results calculated with the ARM method vary from about 8 µT to 13 µT, whereas those with the IRM method range from about 5 µT to 21 µT (Supplementary Discussion 1.2). The high coercivity of sample 392 yields palaeointensities of about 10 µT and 9 µT as calculated with the ARM and IRM methods, respectively, which are consistent with the results of samples 211 and 038 (Fig. 3 and Extended Data Figs. 1 and 3). However, the direction of high coercivity for this sample is scattered and not origin-trending with the MAD < DANG, rendering its palaeointensities less convincing. Therefore, the residual ARM (AREMc) (about <20 µT) and residual IRM (REMc) (about <16 µT) palaeointensities are used for further discussion for sample 392, whereas the ARM- and IRM-corrected palaeointensities are used for the other three samples. The palaeointensity fidelity limit test of the four samples indicates three of them (211, 038 and 392) are ideal recorders and able to record an equivalent thermal remanent magnetization (TRM) field of around 1–7 µT, whereas sample 344 exhibits more erratic behaviour but can nonetheless retrieve a palaeointensity about ≥7–15 µT (Supplementary Table 1 and Supplementary Discussion 1.4). All the recovered palaeointensities of the samples roughly exceed their fidelity limit, demonstrating their reliability. The remanence anisotropy effect on the palaeointensities is about ≤10% and can be neglected for these samples according to the ARM anisotropy results (Supplementary Table 2 and Supplementary Discussion 1.5).

Rock magnetic results indicate that the Chang'e-6 basalt clasts have strongly paramagnetic signals, containing a mixture of magnetic particles with large coercivity distributions and have low

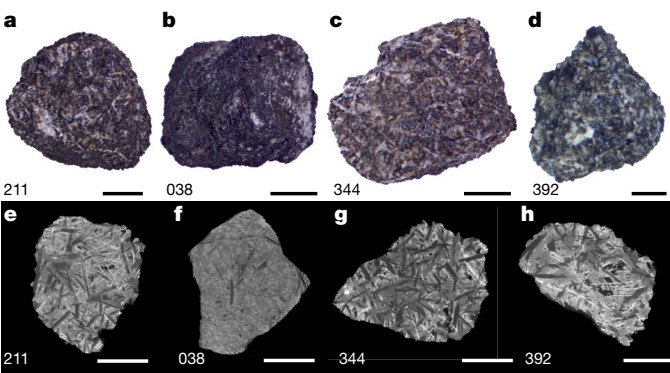

**Fig. 2 | Images of the Chang'e-6 basalt clasts of this study.**
**a**–**d**, Stereomicroscope photos. **e**–**h**, CT transects of the basalt clasts. Scale bars, 2 mm (**a**–**c**,**e**–**g**); 1 mm (**d**,**h**).

magnetic susceptibilities (Extended Data Figs. 4 and 5), which are consistent with the properties of the Chang'e-5 basalts[17]. However, the magnetic-carrying particles of the studied basalts are coarser than those in the Chang'e-5 basalts, except sample 038, which is comparable to the Chang'e-5 samples, as evidenced by the microscopy results showing that µm-sized iron particles are found in samples 211, 344 and 392 (Extended Data Fig. 6), whereas only 100-nm-sized iron particles are found in the Chang'e-5 samples. The rock magnetism and microscopy results consistently demonstrate that the basalt clasts contain stable magnetic carriers of mainly iron particles that are good magnetic recorders of the lunar palaeomagnetic field (Supplementary Discussions 3 and 4).

## Remanence origin of Chang'e-6 basalts

Clarifying the origin of different remanences of the samples is important for deciphering the palaeointensity data. Multiple possible sources of magnetization besides the lunar dynamo, including a crustal magnetic anomaly, VRM and IRM contaminations, and impact remagnetization[6,24] are considered in this study. The magnetic anomaly near the Chang'e-6 landing site at the lunar surface is less than 10 nT according to the global magnetic field model[25] (Fig. 1). Although large discrepancies may exist between the surface field predicted by the orbital data modelling and actual in situ measurements, the maximum lunar surface magnetic anomalies, derived from either orbital predictions or in situ measurements, are hundreds of nanoteslas across the Moon based on our current knowledge[25,26], much lower than the recovered micro-tesla palaeointensities in this study. Forward modelling in the Chang'e-5 landing area estimates an upper limit of an ancient magnetic anomaly to be less than 70 nT (ref. 17), providing a reference that suggests that the ancient magnetic anomaly in the Chang'e-6 landing area is unlikely to exceed this strength considering the scale of volcanism in the Chang'e-6 area is not larger than that of the Chang'e-5 area[27,28]. The palaeointensities recovered from the basalts are about 5–21 µT, which can rule out a magnetic source from a local crustal magnetic anomaly.

The VRM test indicates that the studied basalt clasts are resistant to VRMs with less than 12% of the NRM decaying in 3–8 days in a magnetically shielded room, especially sample 211 with only around 5% decay in about 9 days. The VRM acquisition and decay experiment of sample 392 indicate that the VRM acquired during exposure to the magnetic field of Earth for 2 months unlikely exceeds approximately 5% for the sample (Supplementary Table 3). These results demonstrate that VRMs acquired on Earth can be easily removed, with as low as less than 10 mT of demagnetization according to the AF results (Fig. 3 and Extended Data Figs. 1–3), and thus VRMs are unlikely to contaminate the high coercivity components of the samples. The IRM test indicates that all the NRMs of the samples are lower than the IRMs obtained in a pulse field of 9 mT

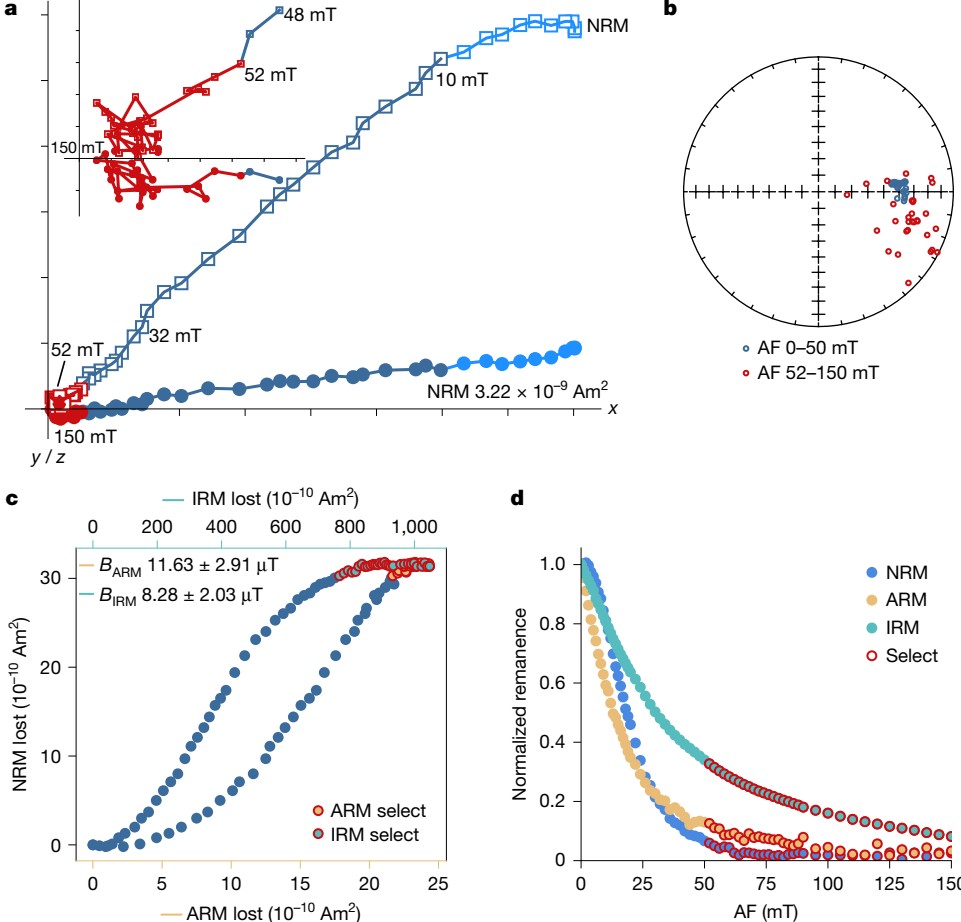

**Fig. 3 | Palaeointensity result of Chang'e-6 basalt clast 211. a**, Orthogonal projection plot of stepwise AF demagnetization. Circles and squares represent the magnetization data projected in the horizontal and vertical planes, respectively. The components of different coercivity are shown in different colours. The high coercivity component is shown in an enlarged inset. **b**, Equal-area projection of the directions during AF demagnetization. **c**, NRM lost versus ARM lost and IRM lost. $B_{ARM}$ and $B_{IRM}$ are the palaeointensities calculated with the ARM and IRM methods, respectively. **d**, NRM, ARM and IRM decay versus AF demagnetization steps. Symbols in red (**a**,**b**) or with red outlines (**c**,**d**) represent the data points used to calculate the palaeointensity.

and that the low-field IRM can be readily removed around the AF steps equivalent to the pulse field when imparting the IRM (Extended Data Fig. 7). These results exclude the possibility that the high coercivity components of the samples were contaminated by a pulse IRM.

Extraterrestrial materials may have potentially experienced complicated influencing histories that may demagnetize the NRM of the sample and/or impart a shock remanent magnetization or TRM from a transient plasma magnetic field on the sample[29,30]. Thus, it is essential to estimate the potential impact history of the sample. We conducted computed tomography (CT), scanning electron microscopy (SEM), Raman spectroscopy and optical microscopy analyses to estimate the extent of impact of the basalt clasts (Methods). The results indicate that the basalt clasts maintain original volcanic mineral crystalline structures with subophitic or porphyritic textures and display no obvious fractures, undulatory extinction or band broadening or shifting of the Raman spectra among the minerals (Fig. 2, Extended Data Figs. 6 and 8, Supplementary Fig. 1 and Supplementary Discussion 4). These results indicate that the samples have experienced limited modification from affecting after their eruption, probably with a peak shock pressure of less than 5 GPa (ref. 31). Moreover, the calculated primary cooling rates of the basalt clasts indicate that they are unlikely to have recorded impact-related transient fields during their original cooling from the magma flows (Supplementary Discussion 1.3).

However, the medium coercivities of the samples, especially sample 211, exhibit peculiar intensities of about 17–80 µT and about 21–139 µT

as calculated by the ARM and IRM methods, respectively. The inconsistent angular differences between the medium coercivity and high coercivity of the samples (Supplementary Table 4) indicate that the medium coercivities were acquired after the fragmentation of the basalts. Their large variations among the palaeointensities, hard and stable behaviour during AF demagnetization (Fig. 3 and Extended Data Fig. 1) and thermal demagnetization behaviour of a sister specimen cut from sample 211, combined with its cooling time simulation results, suggest that the medium coercivities may record a low-temperature TRM acquired from a non-dynamo magnetic source such as an impact-related field[19,32] (Supplementary Discussion 1.3). The results of all these tests indicate that the palaeointensities recorded by the high coercivities of the basalt clasts probably represent the strength of the lunar dynamo magnetic field during the approximately 2.8 Ga volcanic eruption of the Chang'e-6 basalts.

## Variation of the lunar palaeomagnetic field

The previous palaeomagnetic data recovered from returned samples from the Apollo missions show a high-field epoch with Earth-like palaeointensities from 4.2 Ga to 3.5 Ga, followed by a sharp decline to about less than 4 µT around 3.1 Ga (ref. 33). After that, it is suggested that the magnetic field maintained a weak state, probably with a surface strength of several µT, before its demise after 1.0 Ga according to the limited published data[7,13,14,16]. The Chang'e-6 basalt yielding palaeointensities

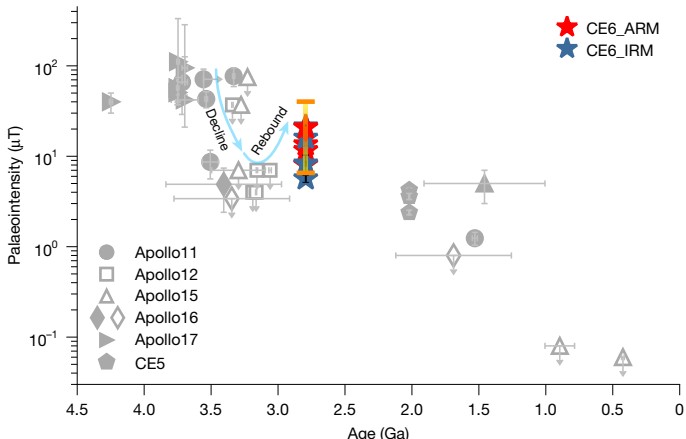

**Fig. 4 | Evolution of the strength of the lunar magnetic field.** The Chang'e-6 basalt palaeointensities argue for a rebound of the lunar dynamo after its first sharp decline around 3.1 Ga. Red and blue stars represent palaeointensities recovered from the Chang'e-6 (CE6) basalt clasts using non-thermal (ARM- and IRM-correction) methods. The orange bar represents the 95% confidence interval (about 7–40 μT with a median value of around 15 μT) of the mean palaeointensities derived from the $10^5$ times resampling from Student's $t$-distribution of the Chang'e-6 palaeointensity data, including both the linear regression error and calibration constants uncertainty (Methods). The Apollo measurements providing only an upper-limit intensity, defined either by the fidelity limit or the AREMc method, are plotted using empty symbols. Data from the Apollo and Chang'e-5 (CE5) missions are compiled from refs. 6,7,14–17,39 and references therein (Supplementary Table 5).

varying from around 5 μT to 21 μT with a median value of about 13 μT at around 2.8 Ga provides a critical anchor for the large gap between 3 Ga and 2 Ga and probably records a rebound of the magnetic field after its first giant decline at around 3.1 Ga despite the large uncertainty of the palaeointensity data (Fig. 4). Although there is the possibility that the Chang'e-6 palaeointensities may overlap with the data between 2 Ga and 1 Ga to some extent if considering the data uncertainties, the results of our statistical analysis indicate the former are highly likely to be stronger than the latter (Supplementary Discussion 6). Therefore, the Apollo lunar palaeointensity compilation (Supplementary Table 5) combined with the data from Chang'e-5 (ref. 17) and now also Chang'e-6 shows that the strength of the lunar dynamo broadly follows a secular trend of an exponential decline over time, which appears linear when converted to log space, albeit with a relatively low correlation coefficient ($r^2 = 0.3$) (Extended Data Fig. 9a,b). We further investigated for the presence of any noticeable second-order fluctuations around this long-term declining trend. Given that the lunar palaeointensity data span multiple orders of magnitude, as well as the exponential trend observed, the logarithm of the raw data is taken before detrending and the linear trend of the data as shown in Extended Data Fig. 9b was used for detrending. The result indicates that following the high-field epoch of 3.9–3.5 Ga (ref. 7), considerable fluctuations of the lunar magnetic field occurred between about 3.5–2.8 Ga (Extended Data Fig. 9c), indicating the lunar dynamo was probably erratic and even episodic.

## Driving mechanism of the lunar dynamo

The palaeointensities recorded by the Chang'e-6 basalt clasts represent the first direct constraint for the palaeomagnetic field on the farside of the Moon. Although no data at about 2.8 Ga are available from the nearside presently, combining with previous data from 3.0 Ga to 1.5 Ga, our results support the existence of a global lunar dynamo, and thus a relatively strong palaeomagnetosphere during this period. Previous studies suggest that the lunar dynamo may have been in a low-power state after its first sharp decline around 3.1 Ga until its demise[13,14].

The proposed protracted weak dynamo is considered to be driven by power sources such as core crystallization or precession. The Chang'e-6 basalt clasts, however, record a rebound of the magnetic field with moderate strong median palaeointensity of about 13 μT around 2.8 Ga, arguing instead that the lunar dynamo was reactivated after its early precipitous decline, powered by either a shifting balance in the dominant power source and/or reinforcement of the original driving mechanism.

When compared with the simulation results of various dynamo models, the Chang'e-6 data align best with the strong surface magnetic field being produced by a basal magma ocean (BMO), which is proposed to have been generated by the emplacement of a radioactive heat-producing and metalliferous layer at the core–mantle boundary during the lunar mantle overturn[34,35] (Fig. 4 and Supplementary Fig. 4). Alternatively, or in combination, the precession dynamo may also serve as a potential magnetic source candidate considering the large uncertainty in the parameters constraining this model[36]. This result indicates that the lunar dynamo was probably driven by a BMO and/or precession at its early-to-mid stage, probably supplemented by other mechanisms such as core crystallization[37]. However, it is noteworthy that there are large uncertainties in the theoretical understanding of the BMO and precession dynamo models[10], and thus the exact lunar dynamo power source remains undetermined and requires further assessment (Supplementary Discussion 5). Based on the updated palaeointensity data now available, the pattern of the lunar magnetic field over time shows its most substantial fluctuations between 3.5 Ga and 2.8 Ga (Extended Data Fig. 9), indicating that the lunar dynamo was strongly variable during this stage, which may serve as a guide for searching for possible magnetic reversals in future lunar exploration missions[6,38].

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

# Article

## Methods

All magnetism-related experiments were conducted at the Palaeomagnetism and Geochronology Laboratory of the Institute of Geology and Geophysics, Chinese Academy of Sciences in Beijing, China. All microscopy analyses were conducted at the Institute of Geology and Geophysics, Chinese Academy of Sciences, and the National Astronomical Observatories, Chinese Academy of Sciences.

### Sample descriptions

Stereomicroscope photos of the basalt clasts show that sample 038 is dark in colour with fine-grained minerals, whereas the other three sample clasts are coarse-grained with large particles (Fig. 2). The basalt clasts are mainly composed of pyroxene, plagioclase, ilmenite and olivine, with minor zircon, apatite and troilite (Extended Data Fig. 6). Micrometre-sized iron grains embedded in troilite crystals were found in samples 211, 344 and 392, whereas iron grains were difficult to observe under the resolution of SEM for sample 038. This result demonstrates that the iron particles in 038 are finer than those in the others, which is similar to the Chang'e-5 basalt clasts[17]. The iron particles appear as inclusions in the rim of pyroxene and thus represent a late phase produced by igneous crystallization[40] as demonstrated in the Chang'e-5 basalts. We dated each studied basalt clast to constrain their crystallization age, which indicates they all belong to the local basalt at the landing site formed in a volcanic eruption dated at about 2,807 ± 3 Ma as constrained by the Pb–Pb data of Zr-bearing minerals and phosphates[8].

### Palaeointensity experiment

The basalt clasts used in this study are small (millimetre-sized), and thus each basalt clast was treated as a single sample without mutually oriented subsamples (Supplementary Discussion 1.3), following the protocol used for Chang'e-5 samples[17]. The samples were fixed in customized quartz holders with background magnetizations of $10^{-12}$ Am$^2$ for palaeointensity analysis[41]. Non-thermal palaeointensity experiments, including the ARM- and IRM-correction methods, were used to recover the ancient intensity of the lunar magnetic field. A sister specimen cut from sample 211 was thermally demagnetized to examine the temperature spectrum of the NRM (Supplementary Discussion 1.3). A fidelity limit test was conducted for each sample to assess if the sample could recover a certain intensity with AF treatment (Supplementary Discussion 1.4). Anisotropy of ARM was also verified to estimate the extent of remanence anisotropy and its effect on the palaeointensity estimation for the studied samples (Supplementary Discussion 1.5).

Remanence measurement, ARM imparting and AF demagnetization were conducted with a 2G RAPID magnetometer (with a sensitivity of $10^{-12}$ Am$^2$) equipped with a d.c. power supply and an AF demagnetizer. IRM was imparted by a pulse magnetizer (MC-1). Thermal treatment was conducted with a magnetic measurement thermal demagnetizer supercooled oven (residual field <10 nT) equipped with a d.c. power supply and an argon purifier system (ZCA-4F). The samples were heated in high-purity (99.999%) argon with an iron sheet as a reducing agent to further reduce oxygen content in the oven.

### VRM and IRM tests

The samples were stored at the National Astronomical Observatories, Chinese Academy of Sciences after their return and exposed to the magnetic field of Earth for about 2 months before being transferred to the magnetically shielded room at the Palaeomagnetism and Geochronology Laboratory. They may have acquired varying degrees of VRM during this time. The NRM decay, VRM acquisition and VRM decay behaviour were investigated to estimate the VRM effect on these samples. The NRMs of the samples were allowed to decay for about 3–9 days in a customized furnace by PYROX with a residual field of less than 10 nT in the magnetically shielded room, and the NRM decay curves were routinely measured. For the VRM decay and acquisition experiment, the sample was placed in the furnace again for 1 week and with an applied stable field of 30 μT (mimicking the ambient field during their storage) to obtain a laboratory VRM. The lab-induced VRM was again decayed in the furnace for 3 days, and the VRM decay curve was measured.

Stepwise IRM acquisition and low-field IRM AF demagnetization measurements were performed to examine if the samples experienced secondary IRM contamination. The samples were first imparted a low-field IRM with the pulse field ranging from 8 mT to 11 mT for the samples. The low-field IRMs were then AF demagnetized until 150 mT with intervals of 1–10 mT. After that, the samples were imparted stepwise IRMs up to 1 T with a pulse magnetizer and the IRM was measured after each step using a 2G RAPID magnetometer.

### Rock magnetism

Magnetic susceptibilities ($\chi$) were measured with an Agico Multifunction Kappabridge susceptibility meter (MFK2-FA). The basalt clasts were fixed in three-dimensional printed resin cylinders designed for irregular small samples[41]. The frequencies used in the measurements were 967 Hz for the low frequency ($\chi_{lf}$) and 15,616 Hz for the high frequency ($\chi_{hf}$) bands, and a field strength of 200 A m$^{-1}$ was used. To minimize the impact of measurement noise, each sample was measured three to five times and an average susceptibility was calculated.

The basalt clasts were placed in nonmagnetic capsules of various sizes, and measurements of hysteresis loops, IRM acquisition, back-field demagnetization curves and first-order reversal curves were conducted with a MicroMag 3900 Vibrating Sample Magnetometer. Hysteresis loops were measured using a discrete sweeping mode with a pausing time of 200 ms, an averaging time of 300 ms and a saturating field of 1 T. The hysteresis data were processed with the program Hystlab v.1.1.1 (ref. 42). The IRM acquisition curves were measured in logarithmic mode ($N_{points} = 120$) over a range from 10 μT to 1 T, with an averaging time of 1 s. The back-field demagnetization curves were measured to acquire the $B_{cr}$ of the samples. The first-order reversal curves of the samples were measured in a discrete sweeping mode, and the pausing time was 200 ms. The maximum field was set to 1 T, with an averaging time of 300 ms. Data were analysed using the FORCinel v3.08 software[43,44] with the smoothing factors $S_{c0} = S_{b0} = 8$, $S_{c1} = S_{b1} = 12$ and $\lambda_h = \lambda_v = 0.1$.

### X-ray CT analysis

To examine bulk microstructures of the samples, X-ray CT analysis was conducted after the palaeointensity experiment with the FEI Heliscan MicroC. The samples were fixed in partially filled plastic tubes and loaded on the scanning stage. Then, they were rotated and moved following a space-filling trajectory, and 2,200–3,010 projections were taken during the measurement. The source voltage was set to 60 kV and the voxel size of the reconstructed images varied from 3.47 μm to 3.97 μm according to the sample size. Data were processed with the Thermo Scientific Avizo software.

### Scanning electron microscopy

Chips from the clasts were mounted in epoxy resin and polished with a grinder to examine the microstructures of the samples. High-resolution backscattered electron images were captured with a field-emission SEM (Zeiss Gemini 450 and Zeiss supra55). Elemental analysis was conducted with energy-dispersive X-ray spectrometer detectors equipped with the SEM. Measurements were performed at an accelerating voltage of 15 kV and a beam current of 2.0–9.0 nA, with a working distance of approximately 8.5 mm.

### Micro-Raman spectroscopy

To estimate the extent of high-temperature and/or high-pressure metamorphism among the samples, Micro-Raman spectroscopy was performed with a Witec alpha300R confocal Raman microscope. The spectra were excited with 532 nm radiation from a semiconductor laser at a power of 1.0–8.0 mW. A 300 grooves per mm grating with a

spectral resolution of 4.8 cm$^{-1}$ was used. The laser beam was focused on the sample surface by a 50/100× Zeiss microscope (NA = 0.75/0.90). A spectral acquisition time of 2–20 s and total spectra with 10–40 accumulations were collected for each measurement.

## Optical microscope analysis

Stereomicroscope photos of the basalt clasts were first taken under reflected light with a stereomicroscope (AOSVI T2-3M180) before any treatment. After all the magnetic analyses, a tiny chip was cut from sample 211 and prepared as a thin section to investigate the peak shock pressure of the sample. The thin section was observed with a Nikon Eclipse LV100N POL microscope, and images were taken under plane-polarized and cross-polarized light.

## Calculation of the 95% confidence interval for palaeointensity

We used a resampling process to calculate the 95% confidence interval of the Chang'e-6 palaeointensities. First, the ARM or IRM palaeointensity of each sample was resampled 10$^5$ times from a Student's $t$-distribution considering regression errors in the palaeointensity estimations. Second, to include uncertainties caused by the calibration constants in the non-heating palaeointensity method, we generated six groups of 10$^5$ calibration factors (three groups of $f'$ for ARM palaeointensity and three groups of $a$ for IRM palaeointensity) using a Gaussian distribution in the exponent assuming a 2 standard deviation factor of 5 for both factors. Then, we multiplied the resampled palaeointensities by the resampled calibration factors to generate six new groups of palaeointensity datasets that include both the linear regression and calibration constant uncertainties. Finally, we randomly selected a datum from each of the six regenerated palaeointensity datasets and calculated the mean palaeointensity. This procedure was repeated 10$^5$ times to generate a dataset containing 10$^5$ mean palaeointensities. The 95% confidence interval of the mean palaeointensities can then be calculated, as shown by an orange error bar in Fig. 4.

## Data availability

Data presented here are provided in the paper, source data, and supplementary materials. The measurement data have been deposited in ScienceDB (https://doi.org/10.57760/sciencedb.16841). Source data are provided with this paper.

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

**Acknowledgements** We thank the Chang'e-6 mission team for returning the lunar samples. The samples studied in this work were allocated by the China National Space Administration. We thank Z. Qiu and H. Tian for their help with the cooling rate calculation of the basalt clasts; J. Gao for assisting in generating lunar magnetic anomaly data; J. Guo and H. Ma for their assistance in sample preparation; S. Liu, Y. Hou and L. Dong for their assistance in laboratory preparation; W. Wu for help with microscope analysis; Y. Lin for discussion of the lunar field variation; Q. Zhou for providing background information on the samples; and R. Mitchell for constructive comments and proofreading. Financial support for this research was provided by the National Natural Science Foundation of China and the Strategy Priority Research Program (Category B) of the Chinese Academy of Sciences (42241101, 42388101, 42488201 and XDB 0710000). S.C. further acknowledges support from the Key Research Program of the Institute of Geology and Geophysics, Chinese Academy of Sciences (IGGCAS-202401).

**Author contributions** Y.P. and R.Z. initiated the project. Y.P. and S.C. conceived and supervised the study. S.C., K.Q., H.Q. and C.D. designed the experiments and analysed the palaeomagnetic data. S.C., K.Q., J.F., P.S., Z.S. and M.Z. conducted the palaeomagnetic measurements. S.Y., C.Z. and X.L. conducted the microscope measurements. K.Q., P.S. and F.C. conducted the simulated calculations. Y.C. and J.L. contributed to the mineralogy discussions. C.L. and H.H. provided background information on the samples. S.C. wrote the paper, with contributions from all authors.

**Competing interests** The authors declare no competing interests.

**Additional information**
**Correspondence and requests for materials** should be addressed to Shuhui Cai.

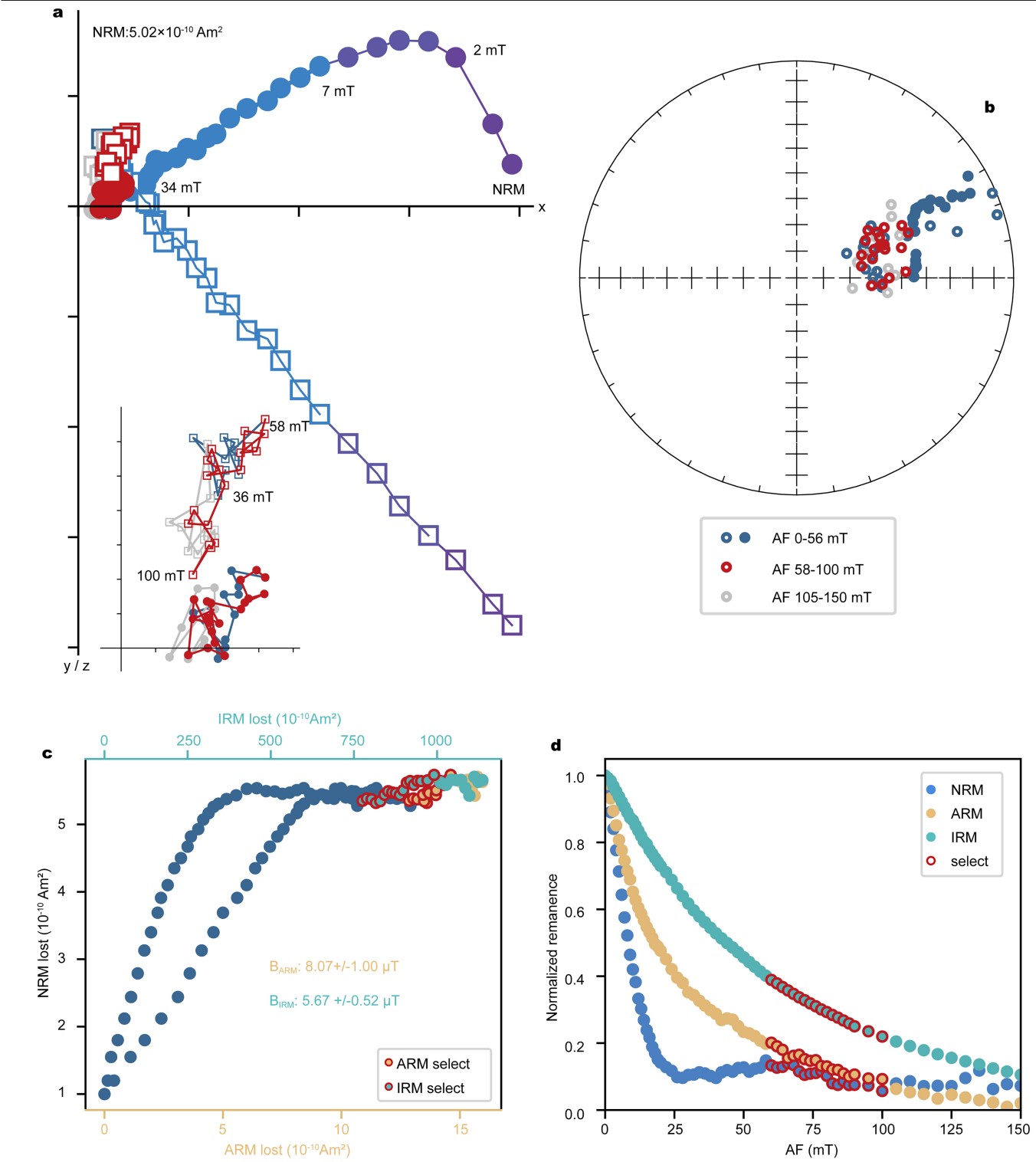

**Extended Data Fig. 1 | Palaeointensity result of the Chang'e-6 basalt clast 038.** Captions are the same as those in Fig. 3.

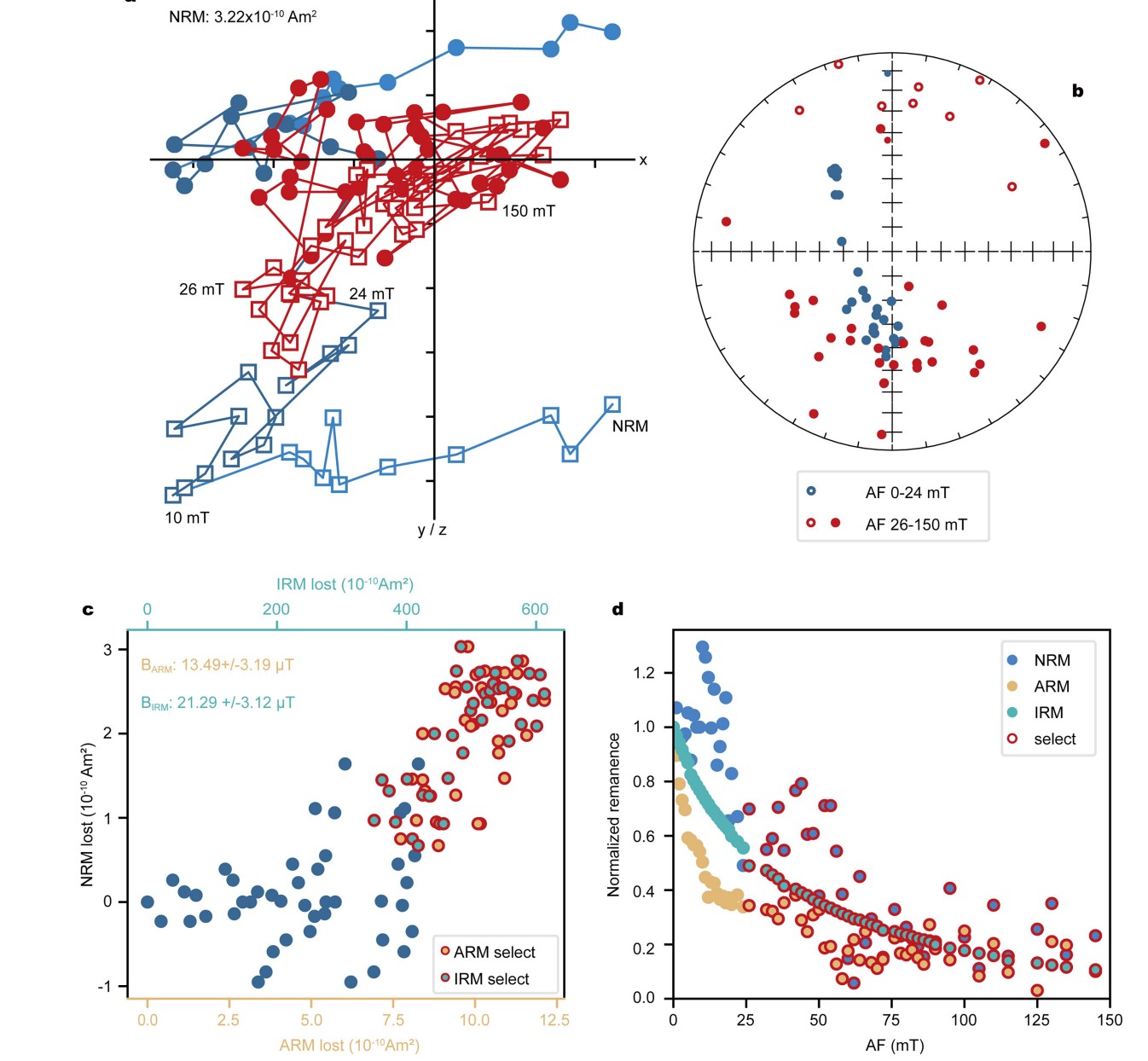

**Extended Data Fig. 2 | Palaeointensity result of the Chang'e-6 basalt clast 344.** Captions are the same as those in Fig. 3.

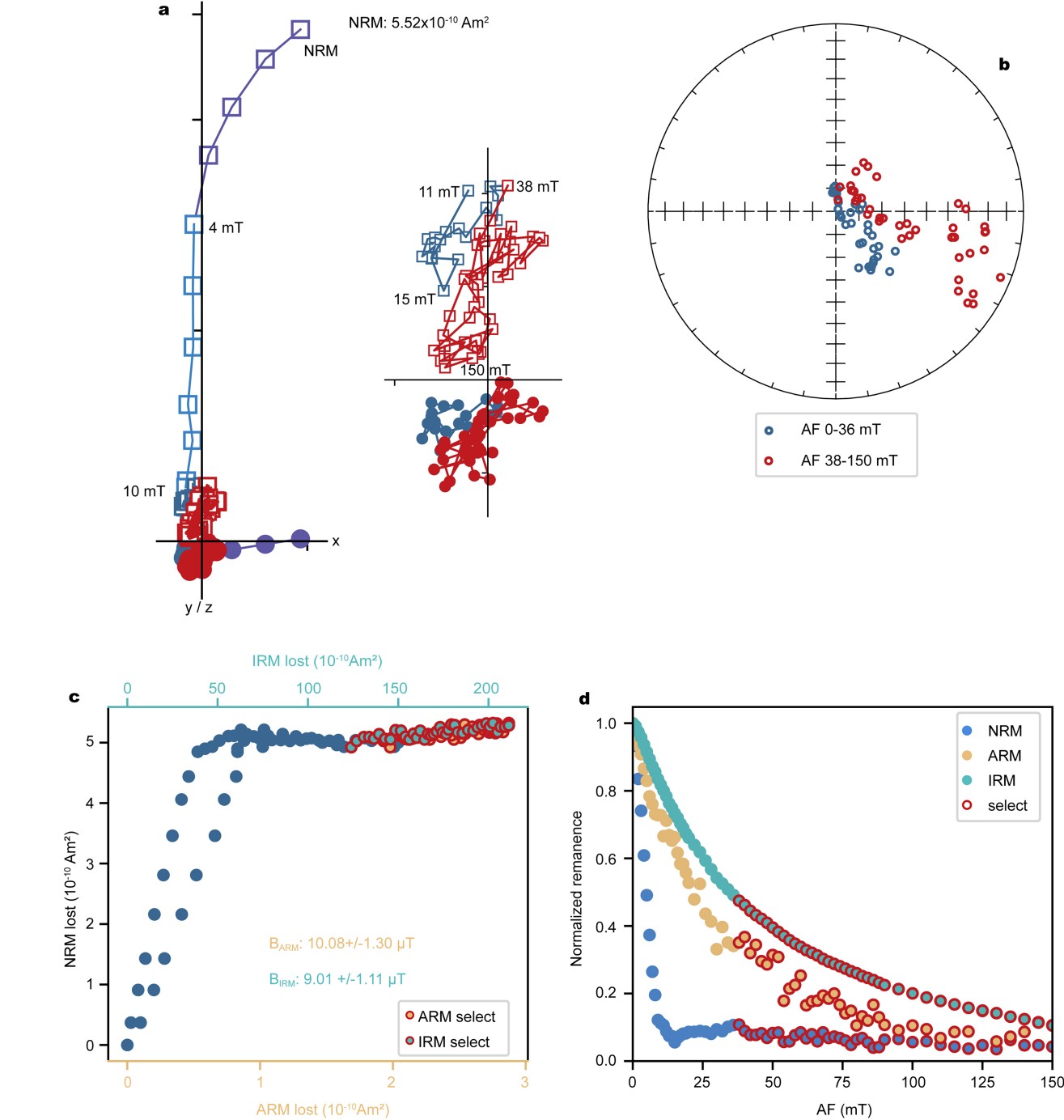

**Extended Data Fig. 3 | Palaeointensity result of the Chang'e-6 basalt clast 392.** Captions are the same as those in Fig. 3.

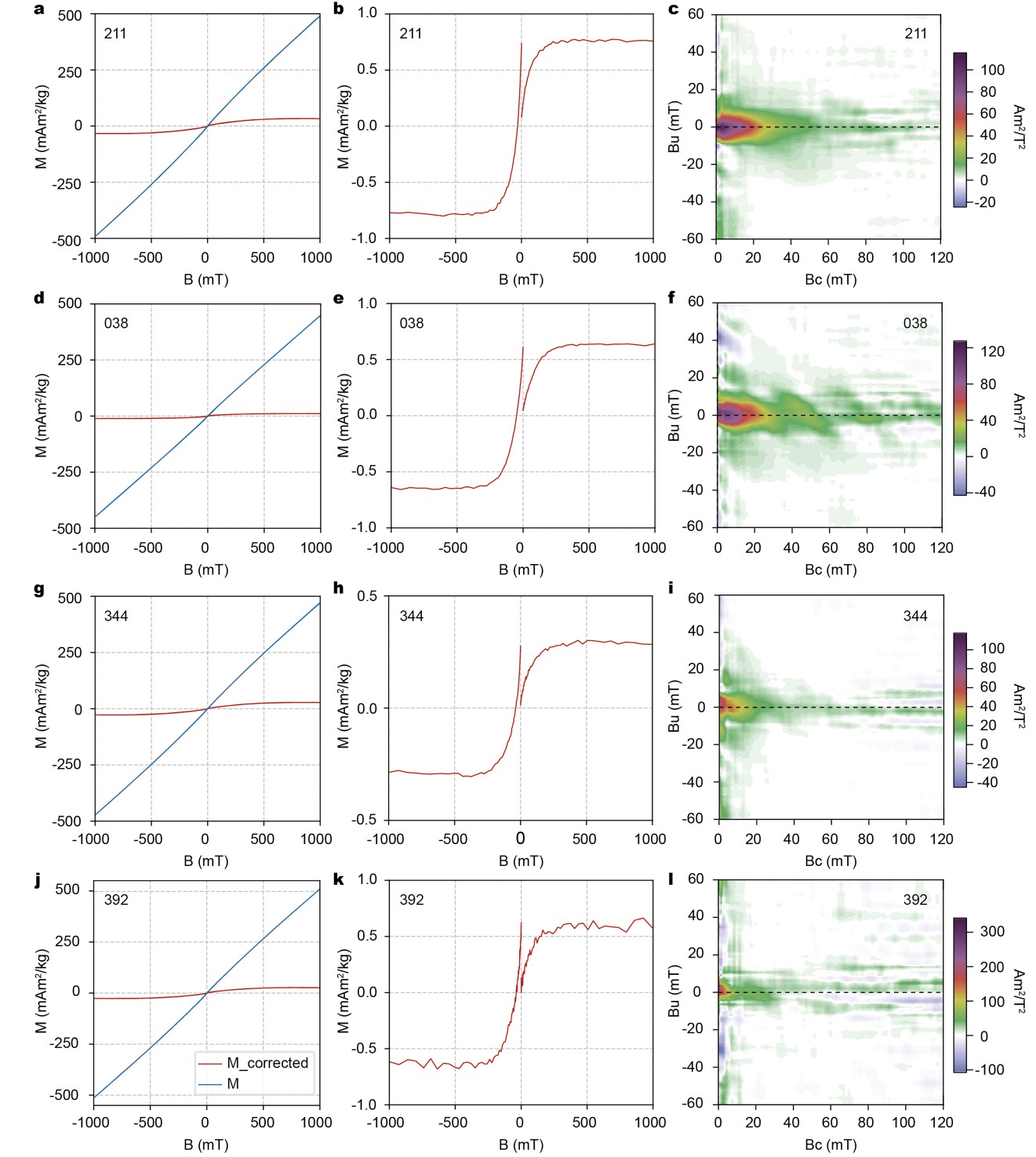

**Extended Data Fig. 4 | Rock magnetic properties of the Chang'e-6 basalt clasts in this study.** The first column (**a**,**d**,**g**,**j**) shows hysteresis loops before (blue) and after (red) paramagnetic corrections. Data were processed with the program HystLab version 1.1.1 (ref. 42). The second column (**b**,**e**,**h**,**k**) shows IRM acquisition and back-field demagnetization curves. The third column (**c**,**f**,**i**,**l**) includes FORC diagrams. Data were processed with the software FORCinel v3.06 (refs. 43,44).

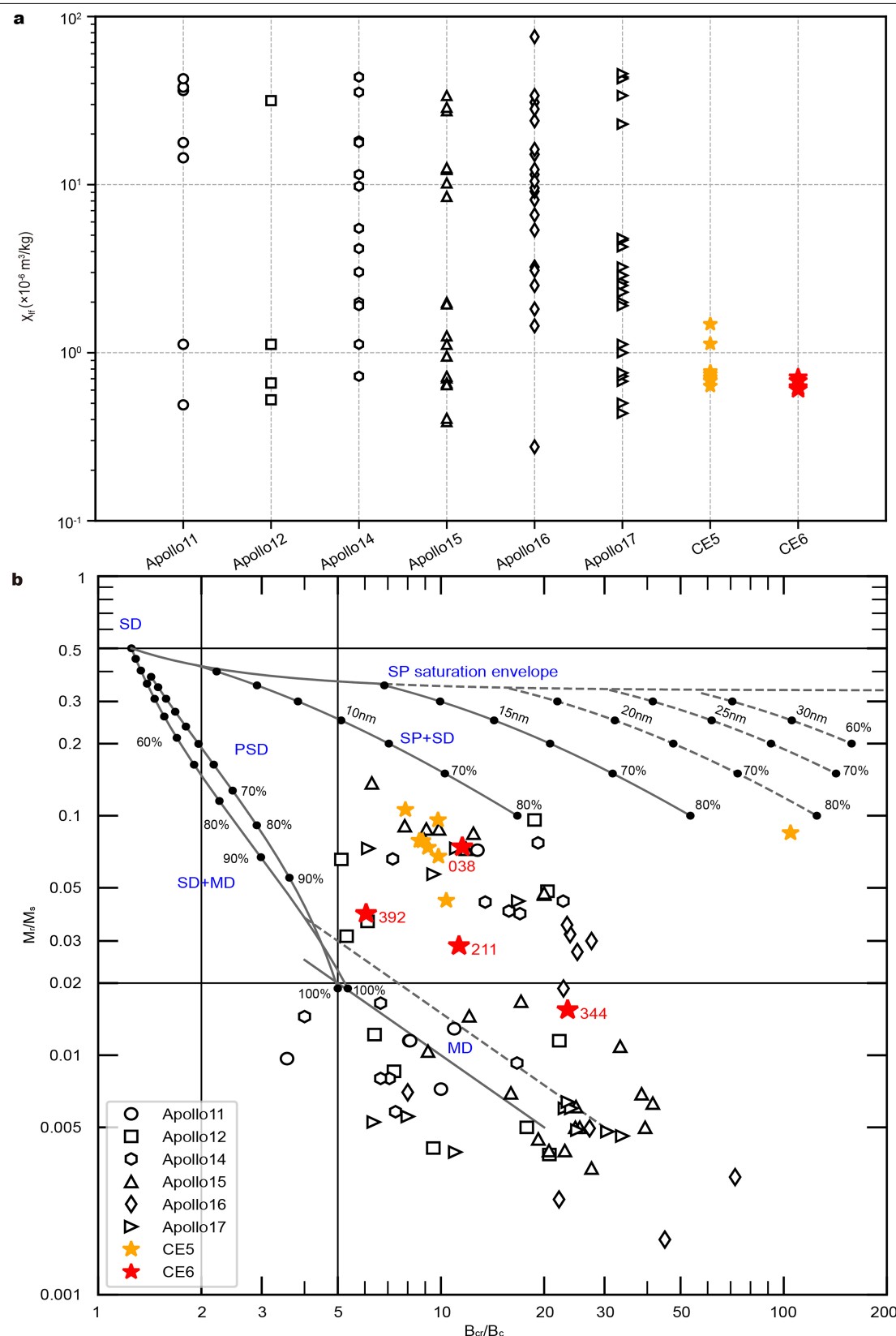

**Extended Data Fig. 5 | Magnetic susceptibility and Day plot of the Chang'e-6 basalt clasts. a**, Low-frequency magnetic susceptibility ($\chi_{lf}$) of the Chang'e-6 samples compared with the Apollo and Chang'e-5 data. **b**, Projection of hysteretic parameters of the Chang'e-6 basalt samples on a Day plot[45]. SP, SD, PSD, and MD represent superparamagnetic, single domain, pseudo-single domain, and multi-domain magnetic particles, respectively. $M_r$, $M_s$, $B_c$, and $B_{cr}$ represent the saturation remanent magnetization, saturation magnetization, coercivity, and remanent coercivity, respectively.

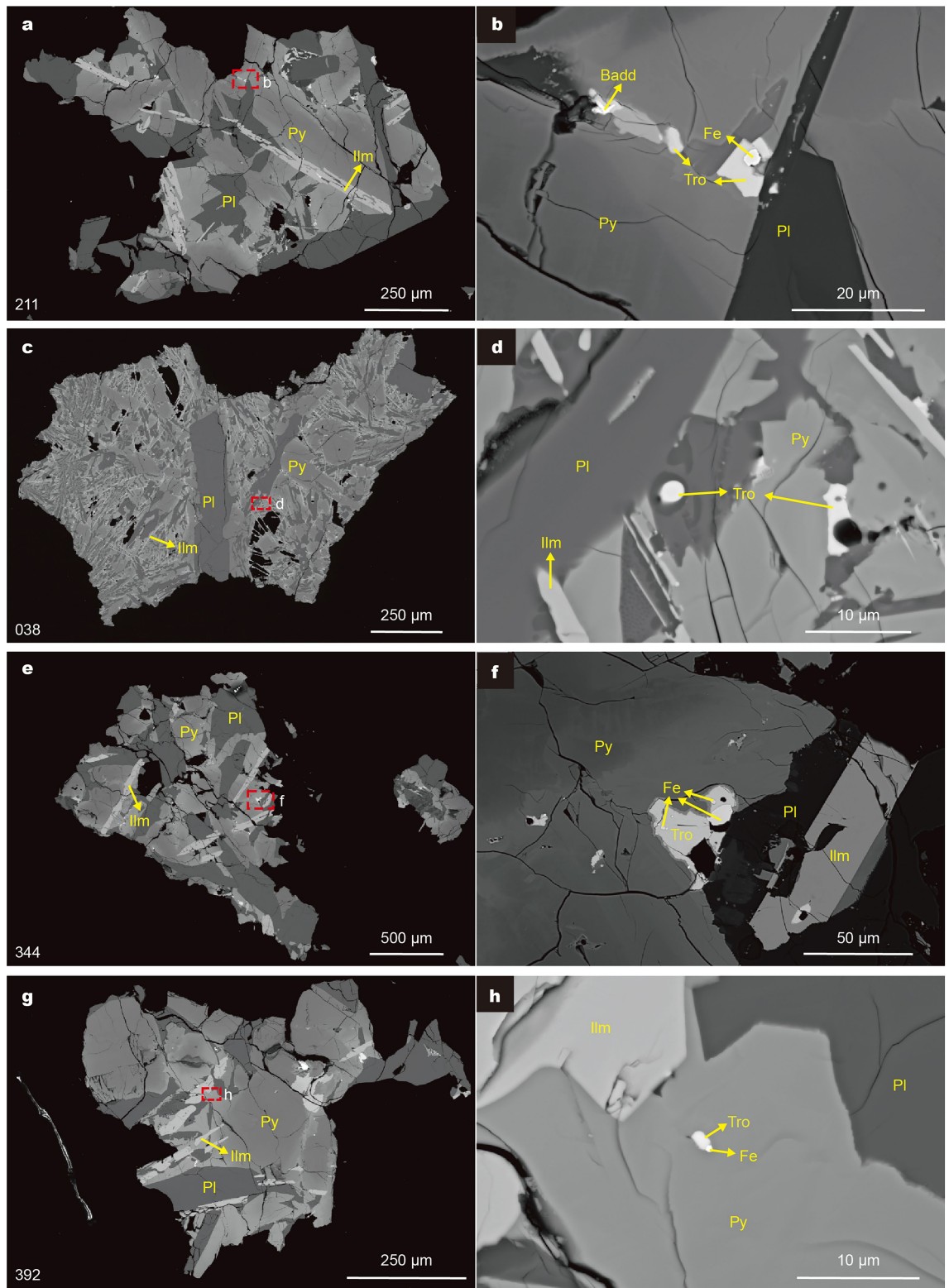

**Extended Data Fig. 6 | Backscattered electron images of the Chang'e-6 basalt clasts in this study.** Plots in the right column (**b**,**d**,**f**,**h**) correspond to the amplified BSE images of the red rectangle areas in the left column (**a**,**c**,**e**,**g**). Badd, baddeleyite; Pl, plagioclase; Py, pyroxene; Ilm, ilmenite; Tro, troilite; Fe, iron.

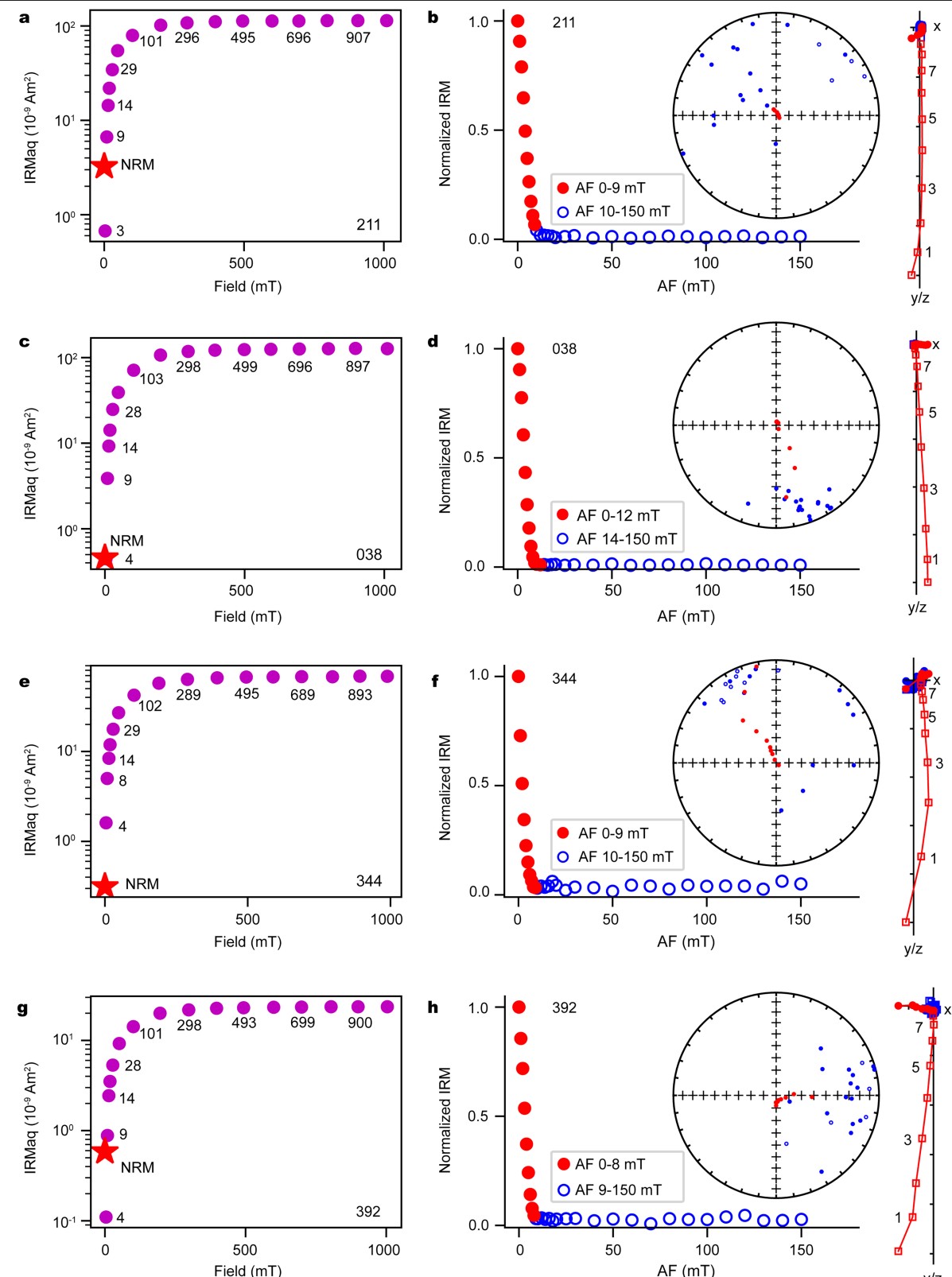

**Extended Data Fig. 7 | IRM test for the Chang'e-6 basalt clasts in this study.** Plots in the left column (**a**,**c**,**e**,**g**) are stepwise IRM acquisitions. Numbers near the circles indicate the pulse field in mT. Red stars are the NRM of the samples measured before AF demagnetization. The right column (**b**,**d**,**f**,**h**) are AF demagnetization results of a low-field IRM of the samples, including the orthogonal projections, equal-area projections, and IRM decay curves. AF steps before and after the pulse field imparting the IRM were denoted by red and blue dots, respectively.

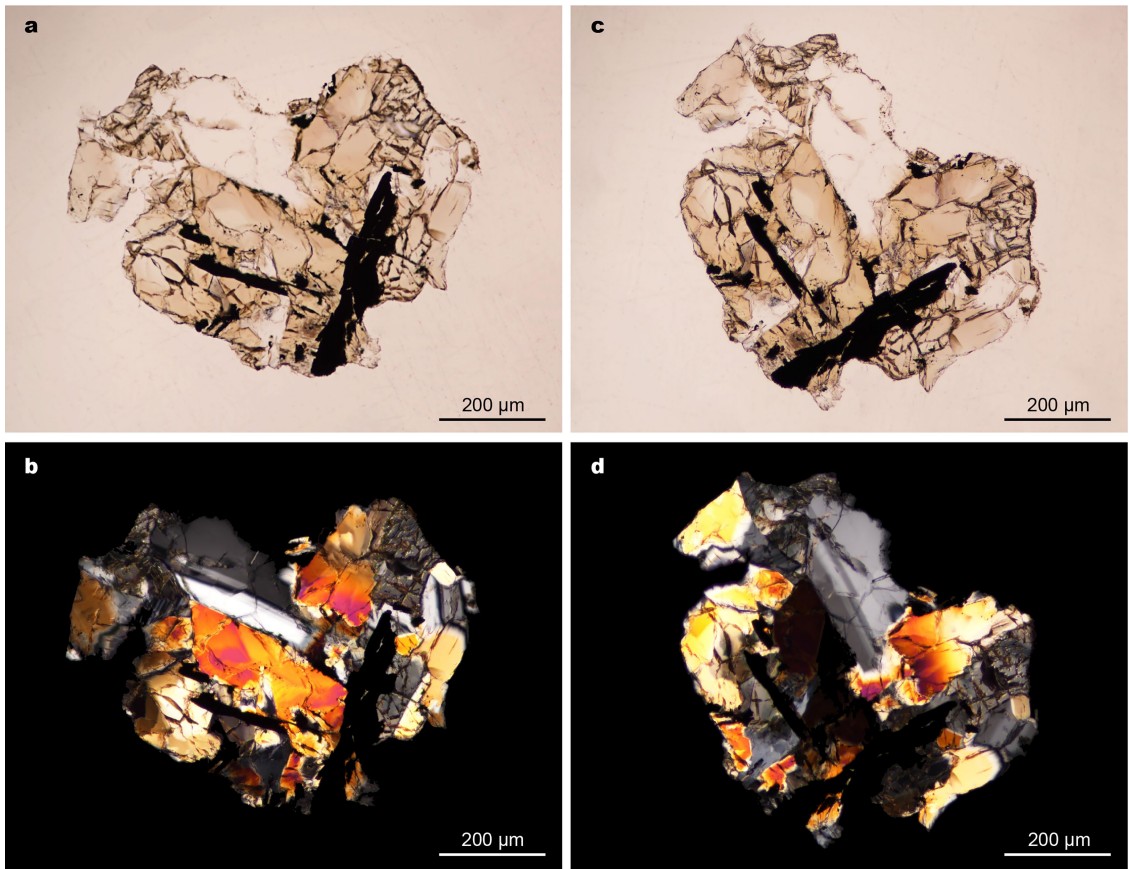

**Extended Data Fig. 8 | Chang'e-6 basalt sample 211 imaged with a polarizing microscope under variable rotation angles. a**,**c**, Images taken under the monopolarizer system. **b**,**d**, Images taken under the crossed polarizer system.

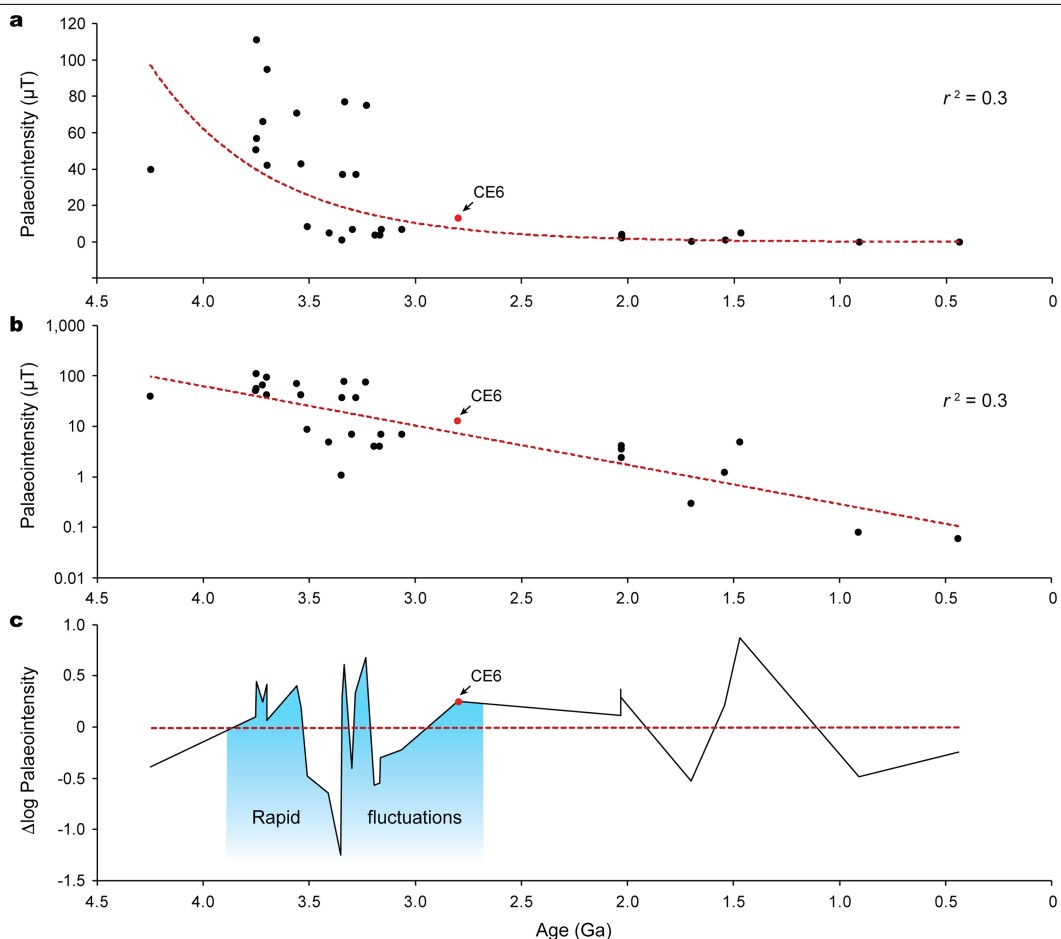

**Extended Data Fig. 9 | Trends and fluctuations in lunar palaeointensity over time.** Lunar palaeointensity data over time in **a** (linear space) and **b** (log space) both fitted with an exponential decay trend. **c**, Logarithm of palaeointensity data detrended (linearly) to exhibit second-order fluctuations, particularly well-sampled and evident between ca. 3.5–2.8 Ga towards the end of which the Chang'e-6 basalts were emplaced.

**Extended Data Table 1 | Palaeointensity data of the Chang'e-6 basalt clasts in this study**

| Sample | Mass (mg) | Method | $B_{ARM}$ (µT) | | | $B_{IRM}$ (µT) | | | MAD | DANG |
|---|---|---|---|---|---|---|---|---|---|---|
| | | | LC | MC | HC | LC | MC | HC | (°) | |
| 211 | 183.3 | ARM, IRM | 25.45 ± 2.81 (0-10 mT) | 80.03 ± 1.04 (10-50 mT) | 11.63 ± 2.91 (52-150 mT) | 95.94 ± 7.71 (0-10 mT) | 139.39 ± 4.50 (10-50 mT) | 8.28 ± 2.03 (52-150 mT) | 30.9 | 8.6 |
| | | AREMc, REMc | | <22.84 | | | <15.36 | | | |
| 038 | 234.9 | ARM, IRM | 21.94 ± 1.07 (0-6 mT) | 16.68 ± 0.51 (7-34 mT) | 8.07 ± 1.00 (58-100 mT) | 62.92 ± 5.82 (0-6 mT) | 20.87 ± 1.31 (7-34 mT) | 5.67 ± 0.52 (58-100 mT) | 25.7 | 11.2 |
| | | AREMc, REMc | | <7.70 | | | <4.06 | | | |
| 344 | 266.6 | ARM, IRM | 26.46 ± 3.06 (0-10 mT) | 37.07 ± 14.89 (10-24 mT) | 13.49 ± 3.19 (26-150 mT) | 78.26 ± 6.27 (0-10 mT) | 54.33 ± 8.09 (10-24 mT) | 21.29 ± 3.12 (26-150 mT) | 25.9 | 15.3 |
| | | AREMc, REMc | | <19.57 | | | <20.09 | | | |
| 392 | 43.7 | ARM, IRM | 220.05 ± 7.35 (0-10 mT) | -5.22 ± 1.55 (11-36 mT) | 10.08 ± 1.30 (38-150 mT) | 363.63 ± 14.49 (0-10 mT) | -5.40 ± 2.22 (11-36 mT) | 9.01 ± 1.11 (38-150 mT) | 22.9 | 60.3 |
| | | AREMc, REMc | | <20.78 | | | <15.96 | | | |

LC, MC, and HC represent low coercivity, medium coercivity, and high coercivity component, respectively. The palaeointensity uncertainty represents 1 standard error resulting from the linear regression process.