## [Peer Review File · Nature]

A reinforced lunar dynamo recorded by Chang'e-6 farside basalt

Corresponding Author: Dr Shuhui Cai

Version 1:

Reviewer comments:

Referee #1

(Remarks to the Author)

This paper reports measurements of the paleomagnetism of basalt particles sampled by the Chang'e-6 mission. These are the first magnetic results from this mission, the first measurements of the past lunar magnetic fields from samples from the lunar far side, and the first measurements from the period 2-3 billion years (Ga) ago. Analyzing 4 mare basalt samples that formed 2.8 billion years (Ga) ago, the authors find that they contain natural remanent magnetization likely dating back to their formation that formed in a paleofield with strength 5-21 uT (15-30% of Earth's present field strength). The authors conclude that a substantial lunar dynamo was present during the moon's mid-life with a field intensity slowly declining until its demise. They argue that the paleointensity history is most likely explained by a basal magma ocean dynamo, but that it may have been supplemented by other mechanisms like precession and core crystallization.

Previous studies of Apollo samples have demonstrated that the Moon once generated a dynamo that lasted from 4.2 to <1 Ga ago. The paleointensity data showed that the field had weakened by orders of magnitude after 3.5 Ga. However, the sparsity of the data mean that it was not clear if this change was abrupt or occurred over billions of years. The possibility that it was abrupt left open the possibility that the dynamo transitioned from an early high-powered mechanism like precession to a late, low-powered one like core convection. There was even recently a pair of studies that boldly proposed there was no lunar dynamo at all after 4.2 billion years ago (although this contradicts both the paleomagnetic record of other Apollo samples as well as spacecraft measurements of lunar crustal magnetization).

This paper presents highly original results that resolve these uncertainties by reporting measurements of the lunar paleofield at both a unique time and place on the Moon not represented by the Apollo and Luna sample suites. In particular, the data fill in a billion-year long gap in the lunar paleomagnetic record and provide the first paleomagnetic measurements from the lunar far side. The study shows that a high intensity field was globally present on the Moon and that lasted until 2.8 Ga, thereby indicating the existence of a high-power, long-lived dynamo mechanism. The authors are to be congratulated on a historic study that provides a major advance in our understanding of lunar magnetism.

Although the samples and results are unique, the study uses a relatively conventional set of paleomagnetic techniques [e.g., alternating field demagnetization, anhysteretic remanent magnetization (ARM) and isothermal remanent magnetization (IRM) paleointensities, paleointensity fidelity tests, hysteresis and first order reversal curves (FORCs), and shock petrography] to measure the magnetization and rock magnetism of the samples. The use of well-established techniques (also used in refs. 2, 5-7, 9-15, 19-21, and 28) means that they are time-tested, such that we've generally established that they work (and know their potential pitfalls).

Inevitably, some readers will likely not accept that paleointensities in this study are robust since they were not measured using thermal demagnetization and remagnetization. Instead, the nonthermal methods used here require the calibration constants f' and a that quantify variations in the ratio of thermoremanence (TRM) to ARM and TRM to saturation IRM (e.g., line 49 of supplementary text). On the other hand, thermal paleointensity studies very often fail due to sample alteration. Furthermore, the available literature on f' and a shows that while their use in nonthermal paleointensity estimates introduces high uncertainties by the standards of earth paleomagnetism studies (2-SD factor of ~5) (Weiss and Tikoo 2014), such uncertainties are usually more than sufficient to test the hypothesis that a dynamo was present on the Moon (but see caveat below about distinguishing the mechanism of such a dynamo). As such, I agree with the main interpretation of these data: three of four samples contain a robust record of the lunar field at 2.8 Ga indicating a paleointensity of a few to several tens of

uT. I also agree that the data likely provide evidence for a dynamo at 2.8 Ga.

With respect to the aforementioned caveat, I am not clear on exactly how the authors estimated the 95% confidence interval on the Chang'e-6 paleointensity mean of 9-16 uT (see caption to figure 4). But this is important because it affects the interpretation of which power source the data support (see below). In particular, did the Student's t test used for this estimate include the uncertainty on each paleointensity datum (Extended Data Table 1)? How were those individual uncertainties estimated? Importantly, do those individual uncertainties take into account the systematic uncertainty in f' and as mentioned above and, if so, what was the value used? Note that the very low uncertainties reported in Extended Table 1 clearly do not take into account the factor of 5 systematic uncertainty on the calibration constants.

The estimation of the uncertainty is important since at the moment, the bottom of the orange bar in Fig. 4 is just a factor of 2 above the field predicts for precession model B2. Nevertheless, this difference apparently leads the authors to favor the basal magma ocean dynamo over precession. But this conclusion might be weakened if the true uncertainty were larger. I note also that independent of the uncertainty on the paleointensity, there are even larger uncertainties in our theoretical understanding of the basal magma ocean and precession dynamos. The conductivity of molten silicate magmas is uncertain by orders of magnitude (e.g., section 1.2 of Scheinberg et al. 2018 EPSL). For the precession dynamo, there are at least two big theoretical uncertainties. First, the evolution of the Moon's orbit is poorly constrained (as demonstrated by the big discrepancies between the 3 different precession models in Fig. 4). Second, and perhaps even more importantly, the expected geometry and vigor of the flows within a precession dynamo are very poorly understood. Therefore, I am not as confident that a basal magma ocean is favored over the precession dynamo. I would suggest changing the last sentence of the abstract to something like: "The result suggest the lunar dynamo was most likely driven by either a basal magma ocean and/or precession, probably supplemented by other mechanisms such as core crystallization."

Other comments:

It is lucky that these samples apparently have much finer metal grain sizes than nearly all Apollo mare basalts, making them unusually good magnetic recorders amongst lunar. This is clearly indicated by the low values for the fidelity limit tests (lines 103-107 and Supplementary Table 1), the coercivity spectra peaking near 100 mT (Supp Fig. 3) and FORC and hysteresis properties showing a substantial single domain grain population (Extended Data Figs. 4, 5). I think it is worth calling out in the text just how unusual the magnetic recording properties of these samples are relative to most other mare basalts. Also, can the authors give a reason for this: are these unusually fast cooled and with fine silicate grain sizes compared to Apollo mare basalts?

The authors conclude that the high coercivity component is a TRM. But if the samples cooled sufficiently fast at 2.8 Ga, it is conceivable they could have recorded a transient field like that proposed to result from impact plasmas. To assess this possibility, I suggest that authors report the primary cooling rate for the samples when they formed as part of mare basalt flow. Grain size and textural analysis can provide such cooling rates (e.g., Shea et al. 2012)

The text does not report the success criteria used for the paleointensity fidelity tests; this should be added. Also, the tests use D and E as metrics. However, I recommend the authors replace the former with the D' metric, which is more robust (see equation (5) of Bryson et al. 2017 EPSL).

Raman spectra can be powerful indicators of shock and it was nice to see the technique used here (Supplementary Figure 1). However, there is no discussion quantitatively linking the peak locations to refractive index values or refractive index values to shock pressures. Without this, it is hard to understand the meaning of the data in this figure. Can the authors add this?

Referee #2

(Remarks to the Author)

The timing and duration of the Moon's magnetic field is a fundamental question in lunar science that has not been resolved after 60 years of study of Apollo samples and subsequent orbital magnetic-field measurements. Most researchers believe that the Moon did have a core dynamo that gave rise to a long-lived global magnetic field. Current controversies are focused on when the dynamo started and stopped, if it was intermittent, how the surface magnetic field strength varied with time, and what energy sources powered the dynamo.

The manuscript "A reinforced lunar dynamo recorded by Chang'e-6 farside basalt" provides new paleomagnetic constraints on the lunar dynamo by analyzing samples collected on the farside of the Moon during the Chang'e-6 mission. The results purport to show moderate field strengths (5-21 micro T) at about 2.8 Ga. These results are important because there are few samples that have been analyzed with ages less than 3.2 Ga, but also because these samples were collected far from the Apollo zone on the equatorial nearside. I believe that these results are important enough to be published in Nature.

I note that I am not an expert in paleomagnetism methods, and I stress the importance of having this manuscript being reviewed by such an expert. I do have some problems with the way these results are interpreted, but these issues, as described below, are easy to rectify and should not pose a hindrance to publication.

Major comments:

1. The authors state on several occasions that the surface field strength was less than about 5 micro T between 1.5 and 1 Ga based on the work of Tikoo et al. (2017). This, however, is an incorrect statement. The work of Tikoo et al. (2017) clearly states that their paleointensity is 5 ± 2 micro T, and not "less than about" 5 micro T.

This error will affect the conclusions of this work because the Tikoo et al. (2017) paleointensity (5 ± 2 micro T) is in fact consistent with the paleointensity found in this study (5-21 micro T). The related manuscript by Cai et al. provides a paleointensity of 2-4 micro T at 2 Ga from the Chang'e-5 samples. Considering the approximate nature of the isothermal-based paleointensity measurements, the different manners in which the uncertainties were calculated, and the different values obtained from ARM and IRM techniques, one could argue that the paleointensity was constant between 2.8 and 1-1.5 Ga. It is my opinion that the authors should acknowledge this possibility in the manuscript, even if it is not their preferred interpretation.

2. The authors state that the Chang'e-6 paleointensity "records a major rebound of the magnetic field strength after its first giant decline [between 3.56 and 3.2 Ga]". I think that this is highly overstated. Between 3.85 and 3.56 Ga there are several paleointensity measurements that are Earth-like, with an average of about 70 micro T (Weiss and Tikoo, 2014). Importantly, these measurements have a lot of variability, with lower bounds being close to 20 micro T. In Tikoo et al. (2014), based on their measurements, they state that the field strength declined to "less than about 4 micro T by 3.19 Ga". Given that the lower bound of the Chang'e-6 paleointensity is 5 micro T, this really is not too different from their results. Yes, there is a 1 micro T difference, and the Tikoo et al. value is an upper bound, but it is important to state these facts accurately as there are multiple ways to interpret what the measurements mean.

So what can we take away from the Chang'e-6 measurements? I would say that they show a dynamo operated after 3.2 Ga. They also show that the paleointensities are lower than the average value of the so-called high-field epoch. They are also consistent with, but perhaps larger than, the Tikoo et al. (2017) paleointensity at 1-1.5 Ga. Given these concerns, I do not believe that it is justified to state that "the strength of the lunar dynamo broadly follows a secular trend of an exponential decline over time, which appears linear when converted to log space". In my opinion, the authors should consider the possibility that the lunar dynamo simply was highly erratic and episodic.

3. The authors state prominently that their paleointensities are most consistent with a model where the dynamo is generated by convection within a basal silicate magma ocean, just above the iron core. Nevertheless, as described in detail in the review of Wieczorek et al. (2023), it is probably not possible to generate a dynamo in a silicate magma ocean. In particular, in order to have a dynamo, the magnetic Reynolds number needs to be larger than a critical value. However, measured electrical conductivities of silicate melts are about 100 times less than required to generate a dynamo. As the Wieczorek et al. review contains many authors that are experts in dynamo modeling, there appears to be little community consensus that the basal magma ocean dynamo of Scheinberg et al. (2018) is viable. At a minimum, the authors should acknowledge this.

A related point concerns plotting the predicted dynamo field strengths of "precession" driven models. There is no scaling relationship that relates the surface field strength to the input power for such dynamos. Given that these dynamos are very different from thermal convection dynamos, involving flow fields generated by parametric resonances, the published models should be treated with extreme caution. The authors of these papers clearly state that when they calculate a field strength that they use a convection-based scaling relationship that might be wrong. Based on the review of Wieczorek et al., it seems that one should probably only note the duration where a precession dynamo could operate, and not give the predicted field strength at all.

4. A recent study by Yang and Wieczorek (2024) showed that some impact craters that are older than 3.9 Ga are magnetized, but that no craters younger than 3.9 Ga are magnetized. Furthermore, they show that a large fraction of the craters younger than 3.9 Ga are demagnetized. One interpretation is that the dynamo field strength was strong before 3.9 Ga, and weak or non-existent afterwards. I think that the authors need to acknowledge that there is an apparent inconsistency between studies based on orbital magnetic field modeling and sample analyses. Those authors give a few explanations for how this difference could be reconciled, one of which is the small number of measurements (from both types of studies) and the likelihood that the dynamo field strength was variable in time. Another possibility is that the ARM and IRM techniques give unreliable paleointensities as advocated by Tarduno et al. (2021).

Minor comments:

1. "Large-scale crustal magnetizations distributed across much of the surface of the Moon (Fig. 1) demonstrate there were once magnetic sources such as lunar dynamo and impact-generated fields that magnetized the lunar crust". This cites Weiss and Tikoo (2014), but much work has been done on lunar crustal magnetization since this time. A more appropriate review would probably be Wieczorek et al. (2023) that was published in the *New Views of the Moon 2* book.

2. Figure 1. It is very difficult to read this figure. Please replot it using a perceptually uniform (scientific) color map. Also, please provide more information about the altitude where the magnetic field is computed. The caption says "surface", but is this really the physical surface (which varies by 20 km in elevation), or is it on a sphere of constant radius (and if so, which

radius was used)? Also, please state explicitly the longitude of the central meridian. Please also provide a citation to the magnetic field model that was used. And lastly, please tell the reader what projection you are using.

3. Figure 4 is very hard to read. Given that the predicted dynamo field strengths are debatable or highly uncertain (see comment above), I would probably just remove these lines and focus on plotting the measurements instead. It is difficult to see what symbols are "empty". Please consider using a different way to plot this. If you do have an interpretation of how the magnetic field strength varies with time (i.e., the exponential model, or a step-like function where the field decreases at a given time), it would probably be appropriate to plot it here.

4. "The magnetic anomaly near the Chang'e-6 landing site at the lunar surface is <10 nT according to the global magnetic field model (Fig. 1)." First, I would probably considering using the Tsunakawa et al. (2015) magnetic field model, as it has a higher spatial resolution than the Ravat et al. (2020) model (as judged by the magnetic field power spectrum). Second, I think that it is dangerous to estimate the magnetic field on the surface at a single point using orbital magnetic field data. Consider two examples. First, at the Apollo 16 landing site, the magnetic field was measured at several places, and its direction was found to change by about 180 degrees (Dyal et al. 1972). Not surprisingly, the orbital magnetic field maps show a value of about zero above the landing site. Second, the predicted field strength at the InSight landing site is about 10 times lower than what was actually measured by the InSight lander (2 micro T, see Johnson et al. 2020). Thus, though I think that the authors should try compare their measured paleointensity to the predicted surface field, they need to emphasize that there are very large problems in doing so. I suspect that pre-existing crustal magnetism is not the cause of the Chang'e-6 rock magnetization, but I am less certain that this is the case for the Chang'e-5 samples.

Version 2:

Reviewer comments:

Referee #1

(Remarks to the Author)

The authors addressed essentially all the major points of my comments as well of those of the other reviewer. I have just one final suggestion, which relates to this statement in the main text:

"The inconsistent angular differences between the MC and HC of the samples (Supplementary Table 4) indicate that the MCs were acquired after the fragmentation of the basalts."

An alternative interpretation of the "inconsistent angular differences" is that both the MC and the HC components were acquired after fragmentation of the basalts. In this case, neither would be a primary record of the lunar field from cooling. Related to this, the authors did not report measurements of mutually oriented subsamples of any one grain, a much more powerful technique for testing whether an HC component is primary and one which has been used extensively in previous paleomagnetic studies. Ideally the authors would provide such an experiments. Otherwise, I suggest I suggest they note explicitly that they were not done and the implications of such.

Referee #2

(Remarks to the Author)

I have read the authors' rebuttal and have re-read the revised manuscript. I feel that the authors have responded to my concerns in good faith and I feel that the manuscript should be published after only addressing some minor concerns that do not require my approval.

1. Abstract "...which attests to an active lunar dynamo at ca. 2.8 Ga in the mid-early stage and argues against the suggestion that the lunar dynamo may have remained in a low-energy state after 3 Ga until its demise." I don't fully understand what the authors mean about a "low energy state". I feel that this sentence could be removed, and instead simply say that this strengthens the case for continuous dynamo activity after about 3.2 Ga and up to its ultimate demise near 1-2 Ga. Another way to look at this is: Perhaps the "low" magnetic field strengths after 3.1 Ga are "normal", and that the high field strengths before 3.1 Ga are a result of "exotic" dynamo mechanisms (like precession).

2. "The result suggests the lunar dynamo was most likely driven by either a basal magma ocean and/or precession, probably supplemented by other mechanisms such as core crystallisation". I do not feel that the results of this study favor any one dynamo process over another. There are simply too many uncertainties involved in the absolute paleointensity of this study (which use isothermal and not thermal techniques). The predicted strength of the various dynamo mechanisms is also highly uncertain. The basal magma ocean dynamo almost certainly can't operate, and the field strength of precession dynamos is unknown.

3. "there were once magnetic sources such as lunar dynamo and impact-generated fields that magnetized the lunar crust". There is little to no evidence for impact generated fields magnetizing the lunar crust (see Oran et al. 2020). I would thus say something like "such as a lunar dynamo, and perhaps impact generated fields".

4. The authors changed the color scale of Figure 1 to the "rainbow" color map. The rainbow colormap is not scientific, and

should never be used in a scientific publication (unless the point is to demonstrate how bad it is!). Please use a perceptually uniform colormap, such as one those presented here: <https://www.fabiocrameri.ch/colourmaps/>. That website provides resources and references demonstrating why one should use perceptually uniform colormaps in science.

5. "Forward modelling in the Chang'e-5 landing area estimates an upper limit of an ancient magnetic anomaly to be <70 nT, providing a reference that suggests that the ancient magnetic anomaly in the Chang'e-6 landing area is unlikely to exceed this strength considering the scale of volcanism in the Chang'e-6 area is not larger than that of the Chang'e-5 area." I personally don't have confidence in these estimates, given that it is known that the field strength can vary dramatically in intensity and direction over short km-scale distances. To convince me these are correct, you would need to predict the field strength at the surface of the Apollo 16 landing site, and compare this prediction with the multiple measurements that were made there. Also, as I mentioned in my review, the field strength at the InSight lander location of Mars is much greater than is predicted based on orbital magnetic field data. Neither of these points were addressed in the revised manuscript.

6. "the strength of the lunar dynamo broadly follows a secular trend of an exponential decline over time, which appears linear when converted to log space, albeit with a relatively low correlation coefficient". In the rebuttal, the authors state that the correlation coefficient is 0.3. I feel that the authors should state this number in the text, because it is so low as to be meaningless. Why not use a linear decline? What is the correlation coefficient of a linear fit?

7. It is a little difficult to understand the difference between the filled and unfilled symbols in Figure 4. Instead of saying "Empty symbols with upper-limit error bars represent intensities....", I would probably say "Measurements that only provide an upper limit are plotted using empty symbols."

8. In the rebuttal, the authors state "The calculated significance value (p) of 0.0 allows us to reject the null hypothesis that these two datasets originate from the same distribution at a confidence level >99%". A p value of 0.0 seems suspiciously small. I would recheck this, and make certain that the statistical technique being used is justified based on the number of data points you have.

Responses in blue

Revisions in red

Line numbers refer to pdf version of the manuscript and Supplementary information file *without* tracked changes.

Editor's comments:

Editorially, your manuscript is already in fairly good shape and we could allow you another 300 words or so in the main text to address the referees' comments. There is also ample room in your Methods section to provide the details requested. If possible, please supply a version of your revised text in MS Word format.

Response: Thanks for all the editor's work and the constructive suggestions.

Revision: We have added and updated some information in the main text according to the referees' comments, which were described in detail in the following response. We have also moved some requested details from the Supplementary information to the Methods section as quoted below. The MS Word format of the revised manuscript and Supplementary information were uploaded to the system.

Added information in the Methods section in the main text:

Lines 338-350: "The NRM of the samples were allowed to decay for ~3–9 days in a customized furnace by PYROX with a residual field <10 nT in a magnetically shielded room, and the NRM decay curves were routinely measured. For the VRM decay and acquisition experiment, sample was placed in the furnace again for one week and with an applied stable field of 30 μ T (mimicking the ambient field during their storage) to obtain a laboratory VRM. The lab-induced VRM was again decayed in the furnace for three days, and the VRM decay curve was measured.

Stepwise IRM acquisition and low-field IRM AF demagnetization measurements were performed to examine if the samples have experienced secondary IRM contamination. The samples were first imparted a low-field IRM with the pulse field ranging from 8–11 mT for the samples. The low-field IRMs were then AF demagnetized until 150 mT with intervals of 1–10 mT. After that, samples were imparted stepwise IRMs up to 1 T with a pulse magnetizer and the IRM was measured after each step using a 2G RAPID magnetometer."

Lines 355-369: "The frequencies used in the measurements were 967 Hz for the low frequency (χ_{lf}) and 15,616 Hz for the high frequency (χ_{hf}) bands, and a field strength of 200 A/m was used. To minimize the impact of measurement noise, each sample was measured 3–5 times and an average susceptibility was calculated.

The basalt clasts were placed in nonmagnetic capsules of various sizes, and measurements of hysteresis loops, IRM acquisition, back-field demagnetization curves, and first-order reversal curves (FORCs) were conducted with a MicroMag 3900 Vibrating Sample Magnetometer. Hysteresis loops were measured using a

discrete sweeping mode with a pausing time of 200 ms, an averaging time of 300 ms, and a saturating field of 1 T. The hysteresis data were processed by the Hystlab version 1.1.1 (ref. 40). IRM acquisition curves were measured in logarithmic mode (Npoints = 120) over a range from 10 μ T to 1 T, with an averaging time of 1 s. The back-field demagnetization curves were measured to acquire the Bcr of the samples. The FORCs of the samples were measured in a discrete sweeping mode, and the pausing time was 200 ms. The maximum field was set to 1 T, with an averaging time of 300 ms. Data were analyzed using the FORCinel v3.08 software^{41,42} with the smoothing factors $Sc0 = Sb0 = 8$, $Sc1 = Sb1 = 12$, and $\lambda_h = \lambda_v = 0.1$.”

Lines 402-415:

“Calculation of the 95% confidence interval for palaeointensity

We employed a resampling process to calculate the 95% confidence interval of the Chang’e-6 palaeointensities. First, the ARM or IRM palaeointensity of each sample was resampled 10^5 times from a Student’s *t*-distribution considering regression errors in the palaeointensity estimations. Secondly, in order to include uncertainties caused by the calibration constants in the non-heating palaeointensity method, we generated six groups of 10^5 calibration factors (three groups of *f*’ for ARM palaeointensity and three groups of *a* for IRM palaeointensity) employing a Gaussian distribution in the exponent assuming a 2 standard deviation factor of 5 for both factors. Then, we multiplied the resampled palaeointensities by the resampled calibration factors to generate six new groups of palaeointensity datasets which include both the linear regression and calibration constant uncertainties. Finally, we randomly selected a datum from each of the six regenerated palaeointensity datasets and calculated the mean palaeointensity. This procedure was repeated 10^5 times to generate a dataset containing 10^5 mean palaeointensities. The 95% confidence interval of the mean palaeointensities can then be calculated, as shown by an orange error bar in Fig. 4.”

Reviewers’ comments:

Referee #1:

This paper reports measurements of the paleomagnetism of basalt particles sampled by the Chang’e-6 mission. These are the first magnetic results from this mission, the first measurements of the past lunar magnetic fields from samples from the lunar far side, and the first measurements from the period 2-3 billion years (Ga) ago. Analyzing 4 mare basalt samples that formed 2.8 billion years (Ga) ago, the authors find that they contain natural remanent magnetization likely dating back to their formation that formed in a paleofield with strength 5-21 μ T (15-30% of Earth’s present field strength). The authors conclude that a substantial lunar dynamo was present during the moon’s mid-life with a field intensity slowly declining until its demise. They argue that the paleointensity history is most likely explained by a basal magma ocean dynamo, but that it may have been

supplemented by other mechanisms like precession and core crystallization.

Previous studies of Apollo samples have demonstrated that the Moon once generated a dynamo that lasted from 4.2 to <1 Ga ago. The paleointensity data showed that the field had weakened by orders of magnitude after 3.5 Ga. However, the sparsity of the data mean that it was not clear if this change was abrupt or occurred over billions of years. The possibility that it was abrupt left open the possibility that the dynamo transitioned from an early high-powered mechanism like precession to a late, low-powered one like core convection. There was even recently a pair of studies that boldly proposed there was no lunar dynamo at all after 4.2 billion years ago (although this contradicts both the paleomagnetic record of other Apollo samples as well as spacecraft measurements of lunar crustal magnetization).

This paper presents highly original results that resolve these uncertainties by reporting measurements of the lunar paleofield at both a unique time and place on the Moon not represented by the Apollo and Luna sample suites. In particular, the data fill in a billion-year long gap in the lunar paleomagnetic record and provide the first paleomagnetic measurements from the lunar far side. The study shows that a high intensity field was globally present on the Moon and that lasted until 2.8 Ga, thereby indicating the existence of a high-power, long-lived dynamo mechanism. The authors are to be congratulated on a historic study that provides a major advance in our understanding of lunar magnetism.

Although the samples and results are unique, the study uses a relatively conventional set of paleomagnetic techniques [e.g., alternating field demagnetization, anhysteretic remanent magnetization (ARM) and isothermal remanent magnetization (IRM) paleointensities, paleointensity fidelity tests, hysteresis and first order reversal curves (FORCs), and shock petrography] to measure the magnetization and rock magnetism of the samples. The use of well-established techniques (also used in refs. 2, 5-7, 9-15, 19-21, and 28) means that they are time-tested, such that we've generally established that they work (and know their potential pitfalls).

Inevitably, some readers will likely not accept that paleointensities in this study are robust since they were not measured using thermal demagnetization and remagnetization. Instead, the nonthermal methods used here require the calibration constants f' and a that quantify variations in the ratio of thermoremanence (TRM) to ARM and TRM to saturation IRM (e.g., line 49 of supplementary text). On the other hand, thermal paleointensity studies very often fail due to sample alteration. Furthermore, the available literature on f' and a shows that while their use in nonthermal paleointensity estimates introduces high uncertainties by the standards of earth paleomagnetism studies (2-SD factor of ~ 5) (Weiss and Tikoo 2014), such uncertainties are

usually more than sufficient to test the hypothesis that a dynamo was present on the Moon (but see caveat below about distinguishing the mechanism of such a dynamo). As such, I agree with the main interpretation of these data: three of four samples contain a robust record of the lunar field at 2.8 Ga indicating a paleointensity of a few to several tens of uT. I also agree that the data likely provide evidence for a dynamo at 2.8 Ga.

The positive comments are appreciated.

With respect to the aforementioned caveat, I am not clear on exactly how the authors estimated the 95% confidence interval on the Chang'e-6 paleointensity mean of 9-16 uT (see caption to figure 4). But this is important because it affects the interpretation of which power source the data support (see below). In particular, did the Student's t test used for this estimate include the uncertainty on each paleointensity datum (Extended Data Table 1)? How were those individual uncertainties estimated? Importantly, do those individual uncertainties take into account the systematic uncertainty in f' and a mentioned above and, if so, what was the value used? Note that the very low uncertainties reported in Extended Table 1 clearly do not take into account the factor of 5 systematic uncertainty on the calibration constants.

Response: The reported uncertainties in Extended Table 1 represent the one standard error associated with the linear regression. In the original text, the 95% confidence interval of the Chang'e-6 palaeointensity mean (orange bar in Fig. 4) was estimated using a resampling method including the linear regression error for each palaeointensity datum. But we do agree with the reviewer that the systematic uncertainty on the calibration constants is important for the interpretation of the driving power source of the lunar dynamo and the variation trend of the lunar paleomagnetic field.

Revision: We originally illustrated uncertainties in Extended Table 1 in Supplementary Discussion 1.2, and now we have also added it to the caption to Extended Data Table 1 for the convenience of readers (Lines 452-453).

“The palaeointensity uncertainty represents 1 standard error resulting from the linear regression process.”

We recalculated the 95% confidence interval considering both the linear regression error and the calibration constants uncertainty. A 2-SD factor of ~ 5 was used for the systematic uncertainty in f' and a as suggested by Weiss and Tikoo (2014), which could represent the maximum range of the Chang'e-6 palaeointensity distribution. The detailed description of the calculation of the 95% confidence interval was updated in Methods section (Lines 402-415).

“Calculation of the 95% confidence interval for palaeointensity

We employed a resampling process to calculate the 95% confidence interval of the Chang'e-6 palaeointensities. First, the ARM or IRM palaeointensity of each sample was resampled 105 times from a Student's t-distribution

considering regression errors in the palaeointensity estimations. Secondly, in order to include uncertainties caused by the calibration constants in the non-heating palaeointensity method, we generated six groups of 105 calibration factors (three groups of f' for ARM palaeointensity and three groups of a for IRM palaeointensity) employing a Gaussian distribution in the exponent assuming a 2 standard deviation factor of 5 for both factors. Then, we multiplied the resampled palaeointensities by the resampled calibration factors to generate six new groups of palaeointensity datasets which include both the linear regression and calibration constant uncertainties. Finally, we randomly selected a datum from each of the six regenerated palaeointensity datasets and calculated the mean palaeointensity. This procedure was repeated 105 times to generate a dataset containing 105 mean palaeointensities. The 95% confidence interval of the mean palaeointensities can then be calculated, as shown by an orange error bar in Fig. 4.”

Fig. 4 was also updated with the new 95% confidence interval (shown below).

Fig. 4 | Evolution of the strength of the lunar magnetic field. The Chang’e-6 basalt palaeointensities argue for a rebound of the lunar dynamo after its first sharp decline around 3.1 Ga. Red and blue stars represent palaeointensities recovered from the Chang’e-6 basalt clasts using non-thermal (ARM- and IRM-correction) methods. The orange bar represents the 95% confidence interval (~7–40 μT with a median value of ~15 μT) of the mean palaeointensities derived from the 10^5 times resampling from Student’s t -distribution of the Chang’e-6 (CE6) palaeointensity data including both the linear regression error and calibration constants uncertainty (Methods). Empty symbols with upper-limit error bars represent intensities of the Apollo samples defined either by the fidelity limit or the AREMc method. Data from the Apollo and Chang’e-5 (CE5) missions are compiled from refs. ^{6,7,13-16,32} (Supplementary Table 5). Shaded colour background indicates the first-order decreasing trend of the strength of the lunar magnetic field over time.

The estimation of the uncertainty is important since at the moment, the bottom of the orange bar in Fig. 4 is just a factor of 2 above the field predicts for precession model B2. Nevertheless, this difference apparently leads the authors to favor the basal magma ocean dynamo over precession. But this conclusion might be weakened if the true uncertainty were larger. I note also that independent of the uncertainty on the paleointensity, there are even larger uncertainties in our theoretical understanding of the basal magma ocean and precession dynamos. The conductivity of molten silicate magmas is uncertain by orders of magnitude (e.g., section 1.2 of Scheinberg et al. 2018 EPSL). For the precession dynamo, there are at least two big theoretical uncertainties. First, the evolution of the Moon's orbit is poorly constrained (as demonstrated by the big discrepancies between the 3 different precession models in Fig. 4). Second, and perhaps even more importantly, the expected geometry and vigor of the flows within a precession dynamo are very poorly understood. Therefore, I am not as confident that a basal magma ocean is favored over the precession dynamo. I would suggest changing the last sentence of the abstract to something like: "The result suggest the lunar dynamo was most likely driven by either a basal magma ocean and/or precession, probably supplemented by other mechanisms such as core crystallization."

Response: Thanks for the advice. We have updated the palaeointensity error according to the reviewer's suggestion. Please see the response to the comments in the preceding response.

Revision: According to the advice, the last sentence of the abstract (Lines 27-29) and the related statement in the main text (Lines 229-231) have both been revised:

"The result suggests the lunar dynamo was most likely driven by either a basal magma ocean and/or precession, probably supplemented by other mechanisms such as core crystallisation."

Please find the details to the main text body revision quoted in our response to major comment 3 of Reviewer 2 below in this document.

Other comments:

It is lucky that these samples apparently have much finer metal grain sizes than nearly all Apollo mare basalts, making them unusually good magnetic recorders amongst lunar . This is clearly indicated by the low values for the fidelity limit tests (lines 103-107 and Supplementary Table 1), the coercivity spectra peaking near 100 mT (Supp Fig. 3) and FORC and hysteresis properties showing a substantial single domain grain population (Extended Data Figs. 4, 5). I think it is worth calling out in the text just how unusual the magnetic recording properties of these samples are relative to most other mare basalts. Also, can the authors give a reason for this: are these unusually fast cooled and with fine silicate grain sizes compared to Apollo mare basalts?

Response: We appreciate the reviewer astutely noting how the Chang'e-6 samples represent ideal magnetic

recorders among lunar samples, and we agree that it is good to draw explicit attention to this. We also agree we must address cooling rates.

Revision: We have calculated the cooling rate of the Chang'e-6 basalt clasts in this study and found that the Chang'e-6 basalt probably cooled faster than some of the reported Apollo mare basalts, which indicates that one possible reason for the difference between the Chang'e-6 and Apollo mare basalts could be the cooling rate differences of the volcanic flows. Faster cooling tends to produce finer grain sizes. We have added a brief discussion following the reviewer's advice, see Supplementary Discussion 3.5 (Lines 221-230). The main text has been slightly revised accordingly (Lines 121-123). The discussion about the cooling rate can be found in the response to the following comment.

Supplementary Discussion 3.5 (Lines 221-230):

“3.5 Magnetic properties of the basalt clasts

Rock magnetic results of the samples, including IRM spectrum analysis, FORCs and projections of the hysteresis parameters on the Day plot, indicate a substantial population of finer SD grains exist in the samples (Supplementary Fig. 3, Extended Data Figs. 4, 5). Combined with the low field values that pass the criteria for the fidelity limit tests (Supplementary Table 1), it indicates that the Chang'e-6 samples have finer metal grain sizes than nearly all Apollo mare basalts, making them good magnetic recorders amongst the lunar samples. The cooling rates calculated by the crystal size of ilmenite in the basalt clasts in this study vary from °C hr⁻¹, higher than that (<3 °C hr⁻¹) of some reported Apollo mare basalts^{2,48}, indicating the Chang'e-6 basalts probably cooled faster and thus produced finer metal grain sizes.”

Main text (Lines 121-123): “The rock magnetism and microscopy results consistently demonstrate that the basalt clasts contain stable magnetic carriers of mainly iron particles that are good magnetic recorders of the lunar palaeomagnetic field (Supplementary Discussion 3-4)”.

The authors conclude that the high coercivity component is a TRM. But if the samples cooled sufficiently fast at 2.8 Ga, it is conceivable they could have recorded a transient field like that proposed to result from impact plasmas. To assess this possibility, I suggest that authors report the primary cooling rate for the samples when they formed as part of mare basalt flow. Grain size and textural analysis can provide such cooling rates (e.g., Shea et al. 2012)

Revision: We have calculated the primary cooling rate for the samples according to the grain size and crystal structures of the ilmenite in the samples using the crystal size distribution (CSD) method. The information was added in Supplementary Discussion 1.3 (Lines 71-81). A sentence was also added in the main text (Lines 159-161).

Supplementary Discussion 1.3 (Lines 71-81):

“The primary cooling rate for the basalt clasts, when they formed as part of a mare basalt flow, is essential for estimating whether the samples could have recorded a transient field such as one resulting from impact plasmas. To assess this possibility, we calculated the primary cooling rate of the basalt clasts in this study according to the grain size and crystal structures of the ilmenite in the samples using the crystal size distribution (CSD) method⁴⁷. The cooling rates of the basalt casts 038, 392, 211, and 344 were estimated to be 106 °C hr⁻¹, 55 °C hr⁻¹, 24 °C hr⁻¹, and 19 °C hr⁻¹, respectively, which probably cooled faster than some of the Apollo mare basalts with reported cooling rates of <3 °C hr⁻¹ (refs. ^{2,48}). However, the results indicate it takes >6 hr, with an average of ~20 hr, for these basalt clasts to cool from the Curie temperature of iron (770°C) to the lunar surface temperature, indicating there is a rather low possibility that the samples, specifically the HC components, have recorded an impact transient field during the original cooling of the volcanic flow(s).”

The text does not report the success criteria used for the paleointensity fidelity tests; this should be added. Also, the tests use D and E as metrics. However, I recommend the authors replace the former with the D' metric, which is more robust (see equation (5) of Bryson et al. 2017 EPSL).

Revision: Thanks for the suggestion. We have replaced D with D' following Bryson et al. (2017) (Supplementary Table 1). The success criteria used for the paleointensity fidelity test were added in the Supplementary text (Lines 125-134 in the Supplementary Discussion 1.4). A related statement was also updated in the main text (Lines 105-107).

Supplementary Table 1 | Statistical parameters of the palaeointensity fidelity limit test of the basalt clasts

Sample	L (μT)	I (μT)	SE (μT)	W (μT)	E (%)	D' (%)	Result	Equivalent TRM field (μT)
211	2	0.88	0.25	1.02	50.88	-56.21	Fail	1.49
211	5	4.12	0.23	0.93	18.60	-17.60	Pass	3.73
211	10	9.74	0.25	1.01	10.14	-2.55	Pass	7.46
038	2	1.72	0.13	0.52	26.20	-13.82	Pass	1.49
038	5	4.37	0.13	0.52	10.40	-12.56	Pass	3.73
038	10	9.76	0.13	0.54	5.42	-2.37	Pass	7.46
344	5	11.84	3.76	15.35	307.07	136.86	Fail	3.73
344	10	34.20	7.32	29.91	299.05	241.96	Fail	7.46
344	20	29.33	4.00	16.35	81.76	46.67	Pass	14.93
344	50	70.47	7.40	30.24	60.48	40.93	Pass	37.31
392	2	3.78	0.97	3.95	197.26	88.86	Fail	1.49

392	5	1.06	0.46	1.87	37.45	-78.89	Fail	3.73
392	10	8.93	0.61	2.50	24.96	-10.69	Pass	7.46
392	20	16.79	1.20	4.90	24.50	-16.03	Pass	14.93

L , laboratory applied DC field; I , retrieved palaeointensity; SE, standard error; W , 95% confidence interval of I ; E , the ratio of W to L ; D' , the relative error between I and L .

Supplementary Discussion 1.4 (Lines 125-134):

“The recovered palaeointensity is compared with the applied DC field (laboratory field) to estimate if the sample has the ability to recover the applied DC field. If a sample fails the test for a certain ARM, then only palaeointensity values larger than the bias field imparting the ARM are considered valid. The revised criteria were suggested by ref. ⁵⁴:

$$E = \frac{W}{L} \quad (1)$$

$$D' = \frac{I - L}{L} \quad (2)$$

where E is the ratio of the 95% confidence interval (W) to the laboratory field (L); D' is the relative error between the retrieved palaeointensity (I) and the laboratory field (L). The test is recognized to ‘Pass’ if D' lies within the interval of (-0.5 and 1) and if E is <0.5 when D' is negative and <1 when D' is positive. Otherwise, the test will be considered to be ‘Fail’.”

Lines 105-107 in the main text: “The palaeointensity fidelity limit test of the four samples indicates three of them (211, 038, and 392) are ideal recorders and able to record an equivalent thermal remanent magnetization (TRM) field of ~1–7 μT ...”

Raman spectra can be powerful indicators of shock and it was nice to see the technique used here (Supplementary Figure 1). However, there is no discussion quantitatively linking the peak locations to refractive index values or refractive index values to shock pressures. Without this, it is hard to understand the meaning of the data in this figure. Can the authors add this?

Revision: We did not measure the refractive index in this study. But we have expanded the interpretation and discussion about the Raman results, including identification and explanation of the peak locations and shapes, how the Raman spectrum responds to shock metamorphism, etc. Please find the details in Supplementary Discussion 4 (Lines 245-259). Supplementary Fig. 1 was also updated with more peak locations marked on the plot.

Supplementary Discussion 4 (Lines 245-259):

“Raman spectra characteristics of the minerals in the samples can be used for detecting the effect of high-temperature and/or high-pressure metamorphism in some cases. For example, the Raman spectrum from

shocked altered plagioclase usually show two broad bands at 480–580 and 900–1,050 cm^{-1} (refs. ⁵⁶⁻⁵⁸). High-temperature and/or high-pressure shock may cause transition of pyroxene to pyroxene glass which may cause broadening of the Raman bands or generation of high-pressure polymorphs such as majorite or akimotoite⁵⁹. The spectrum of plagioclase in the studied basalt clasts is characterized by peaks at 180, 280, 506, and 990 cm^{-1} (Supplementary Fig. 1), which are recognized as the fundamental Raman vibrations of well-crystallized plagioclase. The well-resolved bands at 314–399, 658–666, and 993–1,008 cm^{-1} for pyroxene also indicate that the mineral structure is unaltered by shock-induced metamorphism^{57,60}, which is further supported by the lack of high-pressure polymorphs such as majorite or akimotoite in the samples. Therefore, micro-Raman spectra of the main minerals of plagioclase and pyroxene in the Chang'e-6 basalt clasts indicate the characteristic Raman peaks of the minerals are evident and do not show any obviously disturbed crystallinity such as broadening or shifting, further indicating that the studied clasts did not suffer from obvious impact metamorphism.”

Referee #2:

The timing and duration of the Moon's magnetic field is a fundamental question in lunar science that has not been resolved after 60 years of study of Apollo samples and subsequent orbital magnetic-field measurements. Most researchers believe that the Moon did have a core dynamo that gave rise to a long-lived global magnetic field. Current controversies are focused on when the dynamo started and stopped, if it was intermittent, how the surface magnetic field strength varied with time, and what energy sources powered the dynamo.

The manuscript "A reinforced lunar dynamo recorded by Chang'e-6 farside basalt" provides new paleomagnetic constraints on the lunar dynamo by analyzing samples collected on the farside of the Moon during the Chang'e-6 mission. The results purport to show moderate field strengths (5-21 micro T) at about 2.8 Ga. These results are important because there are few samples that have been analyzed with ages less than 3.2 Ga, but also because these samples were collected far from the Apollo zone on the equatorial nearside. I believe that these results are important enough to be published in Nature.

I note that I am not an expert in paleomagnetism methods, and I stress the importance of having this manuscript being reviewed by such an expert. I do have some problems with the way these results are interpreted, but these issues, as described below, are easy to rectify and should not pose a hindrance to publication.

The positive comments are appreciated.

Major comments:

1. The authors state on several occasions that the surface field strength was less than about 5 micro T between 1.5 and 1 Ga based on the work of Tikoo et al. (2017). This, however, is an incorrect statement. The work of Tikoo et al. (2017) clearly states that their paleointensity is 5 ± 2 micro T, and not "less than about" 5 micro T.

Revision: Sorry for the imprecise statements. We have revised both statements in the main text as follows.

Lines 36-37: "...which first dropped by one order of magnitude around 3.1 Ga, and maintained a low field of several μT until a second decline between ca. 1.5 and 1 Ga".

Lines 178-179: "...it is suggested that the magnetic field maintained a weak state, probably with a surface strength of several μT ".

This error will affect the conclusions of this work because the Tikoo et al. (2017) paleointensity (5 ± 2 micro T) is in fact consistent with the paleointensity found in this study ($5-21$ micro T). The related manuscript by Cai et al. provides a paleointensity of $2-4$ micro T at 2 Ga from the Chang'e-5 samples. Considering the approximate nature of the isothermal-based paleointensity measurements, the different manners in which the uncertainties were calculated, and the different values obtained from ARM and IRM techniques, one could argue that the paleointensity was constant between 2.8 and 1-1.5 Ga. It is my opinion that the authors should acknowledge this possibility in the manuscript, even if it is not their preferred interpretation.

Response: Thanks for pointing out this concern. We agree that there is certain possibility that the Chang'e-6 palaeointensities may overlap with the data between 2 and 1 Ga to some extent if considering the data uncertainties and this is important for explaining the variation trend of the lunar magnetic field after ca. 3 Ga.

Revision: To further assess this issue, we employed a statistical analysis on the Chang'e-6 palaeointensities and published data between 2 and 1 Ga. The statistical analyzing results indicate the Chang'e-6 palaeointensities are highly likely stronger than those between 2 and 1 Ga. Supplementary Discussion 6 was added to describe the details of the test procedure (Lines 297-325):

"6. Kolmogorov-Smirnov test between the Chang'e-6 palaeointensities and data between 2 and 1 Ga

The Chang'e-6 basalt clasts yield palaeointensities varying from $\sim 5-21$ μT with a median value of ~ 13 μT at ca. 2.8 Ga while those published data between 2 and 1 Ga vary from $\sim 0.3-5$ μT if only considering the reported palaeointensity mean values, indicating there is a high possibility that the former recorded stronger palaeointensities than the latter. However, the possibility that these two datasets overlap with each other to some extent still exists if considering the standard error or calibration constant uncertainty of the data. The

variation trend of the lunar magnetic field after ca. 3 Ga thus must be further assessed statistically. The 95% confidence interval of the mean palaeointensities ($\sim 7\text{--}40\ \mu\text{T}$ with a median value of $\sim 15\ \mu\text{T}$) including both the linear regression error and calibration constant uncertainty for the Chang'e-6 basalt clasts were calculated through a resampling process as described in the Methods section. A similar resampling process was conducted on the Chang'e-5 and Apollo data between 2 and 1 Ga (Fig. 4). The data of the Apollo sample (60255) dated at ca. 1.7 Ga was not included because an upper limit was used in the original work¹⁵. In generating each Student's *t*-distribution for the Chang'e-5 data, the ARM and IRM paleointensities of sample 129 were analyzed considering the linear regression error and a 2 standard deviation factor of ~ 5 for the calibration constants. For the non-heating results of sample 118, only the linear regression error was included because its calibration factors were directly derived from the AI-Shaw method¹⁶. For the heating-method result of sample 118 as well as the Apollo results (10018 and 15498) derived from the Thellier-Thellier double-heating technique^{12,14}, only the reported standard error of the palaeointensities was considered.

The resampling datasets for the data between 2 and 1 Ga yielded a 95% confidence interval of $\sim 3\text{--}6\ \mu\text{T}$ with a median value of $\sim 4\ \mu\text{T}$ (Supplementary Fig. 4), which differs significantly from those obtained from the Chang'e-6 data ($\sim 7\text{--}40\ \mu\text{T}$ with a median value of $\sim 15\ \mu\text{T}$). A Kolmogorov-Smirnov statistical test was then performed on the two resampled datasets to further investigate whether they share the same distribution. The calculated significance value (*p*) of 0.0 allows us to reject the null hypothesis that these two datasets originate from the same distribution at a confidence level $>99\%$. Therefore, the statistical analysis indicates that the Chang'e-6 palaeointensities are highly likely to be stronger than those between 2 and 1 Ga.”

A sentence was also added in the main text to make the statement more rigorous (Lines 183-186):

“Although there is the possibility that the Chang'e-6 palaeointensities may overlap with the data between 2 and 1 Ga to some extent if considering the data uncertainties, the results of our statistical analysis indicate the former are highly likely to be stronger than the latter (Supplementary Discussion 6).”

2. The authors state that the Chang'e-6 paleointensity "records a major rebound of the magnetic field strength after its first giant decline [between 3.56 and 3.2 Ga]". I think that this is highly overstated. Between 3.85 and 3.56 Ga there are several paleointensity measurements that are Earth-like, with an average of about 70 micro T (Weiss and Tikoo, 2014). Importantly, these measurements have a lot of variability, with lower bounds being close to 20 micro T. In Tikoo et al. (2014), based on their measurements, they state that the field strength declined to "less than about 4 micro T by 3.19 Ga". Given that the lower bound of the Chang'e-6 paleointensity is 5 micro T, this really is not too different from their results. Yes, there is a 1 micro T difference, and the Tikoo et al. value is an upper bound, but it is important to state these facts accurately as there are multiple ways to interpret what the measurements mean.

Response: Thanks for the comments. We agree that it should be more rigorous to include all these uncertainty descriptions in the text.

Revision: We have added some information in the main text and softened the statement about the rebound of the lunar magnetic field (Lines 180-183):

“The Chang’e-6 basalt yielding palaeointensities varying from ~5–21 μT with a median value of ~13 μT at ca. 2.8 Ga provides a critical anchor for the large gap between 3 and 2 Ga and likely records a rebound of the magnetic field after its first giant decline at ca. 3.1 Ga despite the large uncertainty of the palaeointensity data (Fig. 4).”

So what can we take away from the Chang'e-6 measurements? I would say that they show a dynamo operated after 3.2 Ga. They also show that the paleointensities are lower than the average value of the so-called high-field epoch. They are also consistent with, but perhaps larger than, the Tikoo et al. (2017) paleointensity at 1-1.5 Ga. Given these concerns, I do not believe that it is justified to state that "the strength of the lunar dynamo broadly follows a secular trend of an exponential decline over time, which appears linear when converted to log space". In my opinion, the authors should consider the possibility that the lunar dynamo simply was highly erratic and episodic.

Response: Thanks for the comment. Thanks for the comment. We agree that both the long-term exponential trend in a linear coordinates and the linear relationship in log space of the lunar palaeointensity variation are relatively weak with a small correlation coefficient ($r^2 = 0.3$). This is why we described it as “broadly” following such a trend. But we think it should be more appropriate to point out the large fluctuation and possibility of intermittency of the magnetic field, as the reviewer suggested.

Revision: We revised the statement to “the strength of the lunar dynamo broadly follows a secular trend of an exponential decline over time, which appears linear when converted to log space, albeit with a relatively low correlation coefficient (Extended Data Fig. 9a, b)” (Lines 188-190). We also added a sentence “indicating the lunar dynamo was probably erratic and even episodic” in the main text (Lines 196-197).

3. The authors state prominently that their paleointensities are most consistent with a model where the dynamo is generated by convection within a basal silicate magma ocean, just above the iron core. Nevertheless, as described in detail in the review of Wieczorek et al. (2023), it is probably not possible to generate a dynamo in a silicate magma ocean. In particular, in order to have a dynamo, the magnetic Reynolds number needs to be larger than a critical value. However, measured electrical conductivities of silicate melts are about 100 times less than required to generate a dynamo. As the Wieczorek et al. review contains many authors that are experts in dynamo modeling, there appears to be little community consensus that the basal magma ocean

dynamo of Scheinberg et al. (2018) is viable. At a minimum, the authors should acknowledge this.

A related point concerns plotting the predicted dynamo field strengths of "precession" driven models. There is no scaling relationship that relates the surface field strength to the input power for such dynamos. Given that these dynamos are very different from thermal convection dynamos, involving flow fields generated by parametric resonances, the published models should be treated with extreme caution. The authors of these papers clearly state that when they calculate a field strength that they use a convection-based scaling relationship that might be wrong. Based on the review of Wieczorek et al., it seems that one should probably only note the duration where a precession dynamo could operate, and not give the predicted field strength at all.

Response: Thanks for the comments. We agree that the various lunar dynamo models, especially the basal magma ocean and precession dynamo models, have large uncertainty and limitations, which was also pointed out by Reviewer 1. And thus our new palaeointensity result will give some implication for the lunar dynamo driven mechanism and further assessments are required in future study.

Revision: We have added some discussion about the uncertainty of these models and also softened some of the statements in the text. The predicted lines from various lunar dynamo models in Fig. 4 before were moved to the Supplementary Fig. 4, in line with their deemphasis in revision.

Main text (Lines 224-234):

“When compared to the simulation results of various dynamo models, the Chang’e-6 data align best with the strong surface magnetic field being produced by a basal magma ocean (BMO), which is proposed to have been generated by the emplacement of a radioactive heat-producing and metalliferous layer at the core-mantle boundary during the lunar mantle overturn^{33,36} (Fig. 4, Supplementary Fig. 4). Alternatively, or in combination, the precession dynamo may also serve as a potential magnetic source candidate considering the large uncertainty in the parameters constraining this model³⁴. This result implies that the lunar dynamo was likely driven by a BMO and/or precession at its early-to-mid stage, probably supplemented by other mechanisms such as core crystallisation³⁵. However, it is noteworthy that there are large uncertainties in the theoretical understanding of the BMO and precession dynamo models⁹, and thus the exact lunar dynamo power source remains undetermined and requires further assessment (Supplementary Discussion 5).”

Supplementary Discussion 5 (Lines 273-283):

“But the evolution of the Moon’s orbit and the geometry and vigor of the flows in the precession dynamo model are poorly constrained, and the scaling relationship that relates the surface field strength to the input power is largely unknown⁹, leaving the generated field strength from this dynamo highly variable as demonstrated by the wide field strength ranges between the different precession models in Supplementary Fig.

4. The basal magma ocean (BMO) dynamo model involves the emplacement of a radioactive heat-producing and metalliferous layer at the core-mantle boundary due to the lunar overturn process^{33,36} and could significantly expand the dynamo-generating region. Although the electrical conductivity of molten silicate magmas is uncertain by orders of magnitude³³, by assuming high electrical conductivity in the BMO there is still a possibility that the BMO model achieves a relatively strong surface field of $>10 \mu\text{T}$ (Supplementary Fig. 4).”

4. A recent study by Yang and Wieczorek (2024) showed that some impact craters that are older than 3.9 Ga are magnetized, but that no craters younger than 3.9 Ga are magnetized. Furthermore, they show that a large fraction of the craters younger than 3.9 Ga are demagnetized. One interpretation is that the dynamo field strength was strong before 3.9 Ga, and weak or non-existent afterwards. I think that the authors need to acknowledge that there is an apparent inconsistency between studies based on orbital magnetic field modeling and sample analyses. Those authors give a few explanations for how this difference could be reconciled, one of which is the small number of measurements (from both types of studies) and the likelihood that the dynamo field strength was variable in time. Another possibility is that the ARM and IRM techniques give unreliable paleointensities as advocated by Tarduno et al. (2021).

Response: Thanks for the suggestion. Lunar crustal magnetism and returned sample-based measurements are essential avenues for exploring the lunar magnetic field. But there is some inconsistency between results from the two kinds of analyses, as the reviewer pointed out. There are quite a few possible reasons for this issue, for example, inconsistency between the orbital and *in-situ* measurement anomaly data as the reviewer mentioned in minor comment 4, uncertainties in the palaeointensity data, insufficient measurements from both kind of analyses, and the possibility that the lunar dynamo field was fluctuating greatly and even episodic over time. Additionally, the lunar crust is thought to cool over a long timescales of thousands of years (Weiss et al., 2023; Wieczorek, 2023) and may have recorded multiple ambient magnetic fields, including variations from the lunar dynamo and various impact-related magnetic fields, which could superimpose and cancel each other out. Therefore, it is expected that there are discrepancies in some cases between the orbital lunar magnetic models and palaeomagnetic studies, and thus more reliable measurements are required to further assess this issue.

Revision: We have added some discussion to remind readers about the inconsistency between studies based on orbital magnetic field modelling and sample analyses according to the reviewer’s advice. See lines 43-47 in the main text:

“Therefore, long-standing issues such as the lifetime, geometry, and driving mechanisms of the lunar magnetic field remain greatly debated, where such ambiguity leads to unresolved concerns about inconsistency between

orbital and sample-based measurements data¹⁷ and even permits such contrary perspectives as arguing against the existence of a long-lived lunar core dynamo altogether^{18,19}.”

Minor comments:

1. "Large-scale crustal magnetizations distributed across much of the surface of the Moon (Fig. 1) demonstrate there were once magnetic sources such as lunar dynamo and impact-generated fields that magnetized the lunar crust". This cites Weiss and Tikoo (2014), but much work has been done on lunar crustal magnetization since this time. A more appropriate review would probably be Wieczorek et al. (2023) that was published in the New Views of the Moon 2 book.

Revision: Reference to Wieczorek et al. (2023) has been added as an updated version of the review of lunar magnetism (Line 33).

2. Figure 1. It is very difficult to read this figure. Please replot it using a perceptually uniform (scientific) color map. Also, please provide more information about the altitude where the magnetic field is computed. The caption says "surface", but is this really the physical surface (which varies by 20 km in elevation), or is it on a sphere of constant radius (and if so, which radius was used)? Also, please state explicitly the longitude of the central meridian. Please also provide a citation to the magnetic field model that was used. And lastly, please tell the reader what projection you are using.

Revision: We appreciate the reviewer's constructive feedback and questions on Figure 1. We have replotted Fig. 1 with data generated from the Tsunakawa et al. (2015) magnetic field model instead of the original Ravat et al. (2020) model, considering the reviewer's advice in minor comment 4. The colour was changed to the commonly used rainbow colour scheme. The lunar surface represents a spherical surface of 1,737 km from the centre of the Moon. The longitude of the central meridian is 90°W (chosen for demarcating the lunar nearside/farside on the right/left). The map uses a Winkel Tripel projection. Tsunakawa et al. (2015) is now cited and all this information is now provided in the caption of Fig. 1.

Fig. 1 | Magnetic anomalies on the lunar surface and the landing sites of lunar exploration missions. Magnetic anomaly data are calculated with the lunar magnetic field model of ref. ²⁰. A spherical surface of 1,737 km from the centre of the Moon was used as the lunar surface. The Winkler Tripel projection was used for the map, centred on the lunar nearside/farside boundary at 90°W. The landing sites of the lunar missions are indicated. The South Pole–Aitken basin and Apollo crater are shown as dashed white and yellow lines, respectively.

3. Figure 4 is very hard to read. Given that the predicted dynamo field strengths are debatable or highly uncertain (see comment above), I would probably just remove these lines and focus on plotting the measurements instead. It is difficult to see what symbols are "empty". Please consider using a different way to plot this. If you do have an interpretation of how the magnetic field strength varies with time (i.e., the exponential model, or a step-like function where the field decreases at a given time), it would probably be appropriate to plot it here.

Revision: We have removed the predicted dynamo field lines from Fig. 4 (shown above in this document) and displayed them instead in the newly added Supplementary Fig. 4. The symbols were amplified slightly and the symbol edge lines were plotted thinner to make the 'empty' symbols easier to be distinguished. We have added shaded colour background to indicate the first-order decreasing trend of the strength of the lunar magnetic field. Fig. 4 and its caption have been updated in the main text.

Supplementary Fig. 4 | Predicted strength of the lunar magnetic field from various dynamo models. The basal magma ocean (BMO) models with three different scaling laws used to predict the magnetic field strength are shown: the mixing-length (ML) theory, the balance between Coriolis, inertial, and gravitational (Archimedean) forces (CIA), and the balance between Lorentz, gravitational, and Coriolis forces (magneto-Archimedes-Coriolis, MAC; ref. ³³). Also shown are the precession dynamo models calculated by ref. ³⁴. Models B1 and B2 here are the nominal model in their Figure S3c and model 4 in their Figure S3b, respectively. The core crystallization (CC) dynamo is from the maximum surface value proposed by ref. ³⁵. The orange and red error bars represent the 95% confidence interval of the resampled palaeointensities of Chang’e-6 (CE6) and published data between 2–1 Ga, respectively.

4. "The magnetic anomaly near the Chang'e-6 landing site at the lunar surface is <10 nT according to the global magnetic field model (Fig. 1)." First, I would probably considering using the Tsunakawa et al. (2015) magnetic field model, as it has a higher spatial resolution than the Ravat et al. (2020) model (as judged by the magnetic field power spectrum). Second, I think that it is dangerous to estimate the magnetic field on the surface at a single point using orbital magnetic field data. Consider two examples. First, at the Apollo 16 landing site, the magnetic field was measured at several places, and its direction was found to change by about 180 degrees (Dyal et al. 1972). Not surprisingly, the orbital magnetic field maps show a value of about zero above the landing site. Second, the predicted field strength at the InSight landing site is about 10 times lower than what was actually measured by the InSight lander (2 micro T, see Johnson et al. 2020). Thus, though I think that the authors should try compare their measured paleointensity to the predicted surface field, they need to emphasize that there are very large problems in doing so. I suspect that pre-existing crustal magnetism is not the cause of the Chang'e-6 rock magnetization, but I am less certain that this is the case for the Chang'e-5 samples.

Response: Thanks for the suggestions and comments. We have updated the magnetic field model used to generate the magnetic anomaly data at the lunar surface. The Tsunakawa et al. (2015) model is employed now. Please also see the response to minor comment 2. We agree with the reviewer that large uncertainties may exist among the predicted surface field and it can only be used as a rough reference. Therefore, we also use the forward modelling results for the Chang'e-5 landing area described in the related materials of Cai et al. (2024) to further assess the contribution of the crustal magnetism to the recovered palaeointensity. The estimated ancient magnetic anomaly in the Chang'e-5 landing area is <70 nT, which give us a reference that the ancient magnetic anomaly in the Chang'e-6 landing area is unlikely to exceed this strength considering the scale of volcanism in both areas. And, thus, we considered that the palaeointensities recovered from the CE6 basalts are unlikely to be from a pre-existing local crustal magnetic anomaly as the reviewer possibly suggested.

*****END*****

Responses in blue

Revisions in red

Line numbers refer to version of the manuscript *without* tracked changes.

Editor's comments:

Editorially, your manuscript is generally in good shape and there are only a few issues that we will need for you to attend to (in addition to providing signed forms, publication-quality artwork, etc.). At ~2600 words, your main text is about at our length limit for a short *Nature* Article with four small display items, however we could allow you another 200 words or so to address the referees' remaining comments. Also, please remove the background shading from your Fig. 4.

Response: Thanks for all the editor's work and the constructive suggestions.

Revision:

We have added a few sentences to address the referees' comments, which were described in detail in the following response.

The background shading in Fig. 4 and related figure caption were removed.

Fig. 4 | Evolution of the strength of the lunar magnetic field. The Chang'e-6 basalt palaeointensities argue for a rebound of the lunar dynamo after its first sharp decline around 3.1 Ga. Red and blue stars represent palaeointensities recovered from the Chang'e-6 basalt clasts using non-thermal (ARM- and IRM-correction) methods. The orange bar represents the 95% confidence interval (~7–40 μT with a median value of ~15 μT) of the mean palaeointensities derived from the 10^5 times resampling from Student's t -distribution of the Chang'e-6 (CE6) palaeointensity data including both the linear regression error and calibration constants uncertainty (Methods). The Apollo measurements only providing an upper-limit intensity, defined either by the fidelity limit or the AREM_c method, are plotted using empty symbols. Data from the Apollo and Chang'e-5 (CE5) missions are compiled from refs. ^{6,7,13-16,32} (Supplementary Table 5).

Reviewers' comments:

Referee #1:

The authors addressed essentially all the major points of my comments as well of those of the other reviewer. I have just one final suggestion, which relates to this statement in the main text:

"The inconsistent angular differences between the MC and HC of the samples (Supplementary Table 4) indicate that the MCs were acquired after the fragmentation of the basalts."

An alternative interpretation of the "inconsistent angular differences" is that both the MC and the HC components were acquired after fragmentation of the basalts. In this case, neither would be a primary record of the lunar field from cooling. Related to this, the authors did not report measurements of mutually oriented subsamples of any one grain, a much more powerful technique for testing whether an HC component is primary and one which has been used extensively in previous paleomagnetic studies. Ideally the authors would provide such an experiments. Otherwise, I suggest I suggest they note explicitly that they were not done and the implications of such.

Response: Thanks for the thorough review and constructive comments.

Revision: We have added a statement in the main text and a paragraph of discussion according to the referee's suggestions.

Lines 330-332 in the main text: "The basalt clasts used in this study are small (millimetre-sized), and thus each basalt clast was treated as a single sample without mutually oriented subsamples (Supplementary Discussion 1.3)."

Lines 117-126 in Supplementary Discussion 1.3: "An alternative interpretation of the inconsistent angular differences between MC and HC among the samples could be that both the MC and HC components were acquired after fragmentation of the basalts, and thus neither would be a primary record of the lunar field from cooling. This could be tested by measuring mutually oriented specimens of a sample, which is strongly recommended if applicable. Regrettably, the basalt clasts used in this study are small, at the millimetre scale (Fig. 2), and their NRMs are relatively weak, typically at the order of 10^{-10} Am² (Extended Data Figs. 1–3). Therefore, in order to ensure adequate signal-to-noise ratio, each basalt clast was treated as a single sample without mutually oriented subsamples. However, unlike the MC components, the HC components recorded consistent palaeointensities generally, which almost excludes their possibility of recording an impact-related transient field after fragmentation."

Referee #2:

I have read the authors' rebuttal and have re-read the revised manuscript. I feel that the authors have responded to my concerns in good faith and I feel that the manuscript should be published after only addressing some minor concerns that do not require my approval.

We appreciate for the careful review and constructive comments.

1. Abstract "...which attests to an active lunar dynamo at ca. 2.8 Ga in the mid-early stage and argues against the suggestion that the lunar dynamo may have remained in a

low-energy state after 3 Ga until its demise." I don't fully understand what the authors mean about a "low energy state". I feel that this sentence could be removed, and instead simply say that this strengthens the case for continuous dynamo activity after about 3.2 Ga and up to its ultimate demise near 1-2 Ga. Another way to look at this is: Perhaps the "low" magnetic field strengths after 3.1 Ga are "normal", and that the high field strengths before 3.1 Ga are a result of "exotic" dynamo mechanisms (like precession).

Response: Previous studies suggest that the lunar dynamo may have been in a low-power state after its first sharp decline around 3.1 Ga until its demise as evidenced by their reported low palaeointensities (Tikoo et al., 2017; Weiss et al., 2023; Vervelidou et al., 2023; Cai et al., 2024). One of the most important findings of this work is the recovery of relatively strong palaeointensities at 2.8 Ga, which are generally stronger than those around 3.1 Ga and after 2 Ga, as demonstrated by the statistical analysis (Fig. 4, Supplementary Fig. 4). This result implies a reinforced lunar dynamo at 2.8 Ga, which is contrary to the previous perspectives that the lunar dynamo may have been in a low-power state after its first sharp decline around 3.1 Ga. We appreciate the argument, but consider it more appropriate to retain this sentence.

2. "The result suggests the lunar dynamo was most likely driven by either a basal magma ocean and/or precession, probably supplemented by other mechanisms such as core crystallisation". I do not feel that the results of this study favor any one dynamo process over another. There are simply too many uncertainties involved in the absolute paleointensity of this study (which use isothermal and not thermal techniques). The predicted strength of the various dynamo mechanisms is also highly uncertain. The basal magma ocean dynamo almost certainly can't operate, and the field strength of precession dynamos is unknown.

Response: We agree that both the measurement data and the dynamo mechanisms have uncertainties, which were both considered and discussed in the revised manuscripts. For example, we recalculated the 95% confidence interval of the Chang'e-6 palaeointensities considering both the linear regression error and calibration constants uncertainty (Methods). And the updated 95% confidence interval still support our

conclusion. In order to remind the readers about the uncertainty of the dynamo mechanisms, we explicitly stated in the main text: “However, it is noteworthy that there are large uncertainties in the theoretical understanding of the BMO and precession dynamo models, and thus the exact lunar dynamo power source remains undetermined and requires further assessment”. We also added discussions about the uncertainties of various dynamo mechanisms, especially the basal magma ocean (BMO) and precession, in Supplementary Discussion 5. One of the most challenging aspects about the driving mechanism of the BMO is that it requires high electrical conductivity in the molten silicate magma, which is considered to be feasible during the mid-early stage of the BMO according to mineralogical experiments (Yoshino et al., 2004; Pommier et al., 2024). Therefore, although highly debated, the BMO dynamo could not be entirely excluded. Taking all these together, we think it is probably reasonable to draw some inferences about the driving mechanism of the lunar dynamo at 2.8 Ga based on the current dataset and simulation results. And these inferences certainly need to be verified with new data and theories in the future.

Pommier, A., Walter, M. J., Hao, M., Yang, J. & Hrubciak, R. Acoustic and electrical properties of Fe-Ti oxides with application to the deep lunar mantle. *Earth Planet. Sci. Lett.* 628 (2024).

Yoshino, T., Walter, M. J. & Katsura, T. Connectivity of molten Fe alloy in peridotite based on in situ electrical conductivity measurements: implications for core formation in terrestrial planets. *Earth Planet. Sci. Lett.* 222, 625-643 (2004).

Revisions: We revised the sentence in Supplementary Discussion 5 (Lines 291-295) to include a statement about the electrical conductivity of the molten silicate magma: “Although the electrical conductivity of molten silicate magmas is uncertain by orders of magnitude³⁵, by assuming high electrical conductivity in the BMO which is considered to be feasible during the mid-early stage of the BMO according to mineralogical experiments^{65,66}, there is still a possibility that the BMO model achieves a relatively strong surface field of >10 μ T (Supplementary Fig. 4).”

3. "there were once magnetic sources such as lunar dynamo and impact-generated fields that magnetized the lunar crust". There is little to no evidence for impact generated

fields magnetizing the lunar crust (see Oran et al. 2020). I would thus say something like "such as a lunar dynamo, and perhaps impact generated fields".

Thanks for the suggestion.

Revision: Revised. Lines 32-33 in the main text: "...there were once magnetic sources such as lunar dynamo, and perhaps impact-generated fields that magnetized the lunar crust."

4. The authors changed the color scale of Figure 1 to the "rainbow" color map. The rainbow colormap is not scientific, and should never be used in a scientific publication (unless the point is to demonstrate how bad it is!). Please use a perceptually uniform colormap, such as one those presented here: <https://www.fabiocrameri.ch/colourmaps/>. That website provides resources and references demonstrating why one should use perceptually uniform colormaps in science.

Thanks for the suggestion.

Revision: Fig. 1 was revised and "Roma" color map is used now.

5. "Forward modelling in the Chang'e-5 landing area estimates an upper limit of an ancient magnetic anomaly to be <70 nT, providing a reference that suggests that the ancient magnetic anomaly in the Chang'e-6 landing area is unlikely to exceed this strength considering the scale of volcanism in the Chang'e-6 area is not larger than that of the Chang'e-5 area." I personally don't have confidence in these estimates, given that

it is known that the field strength can vary dramatically in intensity and direction over short km-scale distances. To convince me these are correct, you would need to predict the field strength at the surface of the Apollo 16 landing site, and compare this prediction with the multiple measurements that were made there. Also, as I mentioned in my review, the field strength at the InSight lander location of Mars is much greater than is predicted based on orbital magnetic field data. Neither of these points were addressed in the revised manuscript.

Response: We agree that the field strength can vary largely in intensity and direction over short km-scale distances. The forward modelling in Cai et al. (2024) considered the stratigraphic structures (such as thickness of the basalt layer and lunar crust) in the Chang'e-5 landing area, used the saturation remanent magnetization (M_r), and assumed two magnetization directions in the model to estimate an upper limit for the ancient magnetic anomaly at the lunar surface. And we think the results can give a reference for the ancient magnetic anomaly in the Chang'e-6 landing area because one of the main factors controlling the strength of the magnetic anomaly is the scale of the basalt in the landing area. There are indeed many assumptions incorporated in the calculation and thus there must be uncertainties. But it can at least give an upper-limit estimation. The other question the referee emphasized is the discrepancy between predictions from orbital magnetic field data and *in-situ* measurements. We totally agree that there must be discrepancies between them. However, we do not think it affects our main conclusion because based on our current knowledge, the estimated maximum magnetic anomaly at the lunar surface from both the orbital predictions and *in-situ* measurements is at the level of hundreds of nanoteslas across the Moon, which is far lower than our measurement value of $\sim 5\text{--}21\ \mu\text{T}$. Therefore, we do not consider lunar crustal magnetization could be the magnetic source of the Chang'e-6 palaeointensity, as also mentioned by the referee in the first round of comments.

Revision: In order to remind the reader about all kinds of uncertainties as the referee suggested, we added some statements in the main text (Lines 111-114):

“Although large discrepancies may exist between the surface field predicted by the orbital data modelling and actual *in-situ* measurements, the maximum lunar surface

magnetic anomalies, derived from either orbital predictions or *in-situ* measurements, are hundreds of nanoteslas across the Moon based on our current knowledge^{21,26}, much lower than the recovered micro-tesla palaeointensities in this study.”

6. "the strength of the lunar dynamo broadly follows a secular trend of an exponential decline over time, which appears linear when converted to log space, albeit with a relatively low correlation coefficient". In the rebuttal, the authors state that the correlation coefficient is 0.3. I feel that the authors should state this number in the text, because it is so low as to be meaningless. Why not use a linear decline? What is the correlation coefficient of a linear fit?

Response: We appreciate the reviewer's request to see a linear fit, but we would still argue for an exponential trend for the raw data that span multiple orders of magnitude and flatten out after ca. 2 Ga. Firstly, a linear trend yields an equally weak correlation ($r^2 = \sim 0.3$). Secondly, the residuals for the linear fit would imply that the two youngest data would represent a stronger field than the long-term trend (see the plot below), which is inconsistent with the generally accepted notion of a declining dynamo by that late stage. Thirdly, whether a linear or exponential trend is preferred, the presence of strong second-order fluctuations, with CE6 representing an increase after the ca. 3 Ga low, is still there. Finally, we should note that part of the main reason the correlation coefficients of the long-term trends are so low is, at least in large part, due to the large magnitude of the second-order fluctuations we emphasize in this paper. We thank the reviewer for asking us to look into this issue further.

Linear fit result of the lunar palaeointensity data.

Revision: The coefficient 0.3 was stated in the text (Line 175 in the main text).

7. It is a little difficult to understand the difference between the filled and unfilled symbols in Figure 4. Instead of saying "Empty symbols with upper-limit error bars represent intensities....", I would probably say "Measurements that only provide an upper limit are plotted using empty symbols."

Thanks for the suggestion.

Revision: Revised. Lines 306-307 in the main text: "The Apollo measurements only providing an upper-limit intensity, defined either by the fidelity limit or the AREMc method, are plotted using empty symbols."

8. In the rebuttal, the authors state "The calculated significance value (p) of 0.0 allows us to reject the null hypothesis that these two datasets originate from the same distribution at a confidence level >99%". A p value of 0.0 seems suspiciously small. I would recheck this, and make certain that the statistical technique being used is justified based on the number of data points you have.

Response: Thanks for the reminding. We have rechecked the calculation and did not find any problem. The Kolmogorov-Smirnov test is a widely used statistical technique to test if two datasets are from the same distribution, which was also used to constrain the decline of the lunar magnetic field by Tikoo et al. (2014). We employed a resampling (10^5 times) method to reduce the impact of insufficient data. The 95% confidence intervals of the two datasets ($\sim 7\text{--}40\ \mu\text{T}$ for Chang'e-6 and $\sim 3\text{--}6\ \mu\text{T}$ for data from 2–1 Ga) do not overlap, which indicates they are statistically different. The Kolmogorov-Smirnov test further demonstrates they are from different distributions at a confidence level >99%.

Tikoo, S. M. et al. Decline of the lunar core dynamo. *Earth Planet. Sci. Lett.* 404, 89-97 (2014).